



# Leveraging the signature of heterotrophic respiration on atmospheric CO$_2$ for model benchmarking

**Samantha J. Basile**[1]**, Xin Lin**[1]**, William R. Wieder**[2,3]**, Melannie D. Hartman**[2,4]**, and Gretchen Keppel-Aleks**[1]

[1]Department of Climate and Space Sciences and Engineering, University of Michigan, Ann Arbor, MI 48105, USA
[2]Climate and Global Dynamics Laboratory, National Center for Atmospheric Research, Boulder, CO 80305, USA
[3]Institute of Arctic and Alpine Research, University of Colorado, Boulder, CO 80309, USA
[4]Natural Resource Ecology Laboratory, Colorado State University, Fort Collins, CO 80523, USA

**Correspondence:** Samantha J. Basile (sjbasile@umich.edu)

**Abstract.** Spatial and temporal variations in atmospheric carbon dioxide (CO$_2$) reflect large-scale net carbon exchange between the atmosphere and terrestrial ecosystems. Soil heterotrophic respiration (HR) is one of the component fluxes that drive this net exchange, but, given observational limitations, it is difficult to quantify this flux or to evaluate global-scale model simulations thereof. Here, we show that atmospheric CO$_2$ can provide a useful constraint on large-scale patterns of soil heterotrophic respiration. We analyze three soil model configurations (CASA-CNP, MIMICS, and CORPSE) that simulate HR fluxes within a biogeochemical test bed that provides each model with identical net primary productivity (NPP) and climate forcings. We subsequently quantify the effects of variation in simulated terrestrial carbon fluxes (NPP and HR from the three soil test-bed models) on atmospheric CO$_2$ distributions using a three-dimensional atmospheric tracer transport model. Our results show that atmospheric CO$_2$ observations can be used to identify deficiencies in model simulations of the seasonal cycle and interannual variability in HR relative to NPP. In particular, the two models that explicitly simulated microbial processes (MIMICS and CORPSE) were more variable than observations at interannual timescales and showed a stronger-than-observed temperature sensitivity. Our results prompt future research directions to use atmospheric CO$_2$, in combination with additional constraints on terrestrial productivity or soil carbon stocks, for evaluating HR fluxes.

# 1 Introduction

Atmospheric CO$_2$ observations reflect net exchange of carbon between the land and oceans with the atmosphere. Observations of atmospheric CO$_2$ concentration have been collected in situ since the late 1950s (Keeling et al., 2011), and global satellite observations have become available within the last decade (Crisp et al., 2017; Yokota et al., 2009). The high precision and accuracy of in situ observations and the fact that these measurements integrate information about ecosystem carbon fluxes over a large spatial footprint make atmospheric CO$_2$ a strong constraint on model predictions of net carbon exchange (Keppel-Aleks et al., 2013). For example, at seasonal timescales, atmospheric CO$_2$ can be used to evaluate the growing-season net flux, especially in the Northern Hemisphere (Yang et al., 2007). At interannual timescales, variations in the atmospheric CO$_2$ growth rate are primarily driven by changes in terrestrial carbon fluxes in response to climate variability (Cox et al., 2013; Humphrey et al., 2018; Keppel-Aleks et al., 2014). Recent studies have hypothesized that soil carbon processes represent one of the key processes in driving these interannual variations (Cox et al., 2013; Wunch et al., 2013). Moreover, soil carbon processes represent one of the largest uncertainties in predicting future carbon–climate feedbacks, in part because non-permafrost soils contain an estimated 1500 to 2400 PgC (Bruhwiler et al., 2018), at least a factor of 3 larger than the preindustrial atmospheric carbon reservoir.

Soil heterotrophic respiration (HR), the combination of litter decay and microbial breakdown of organic matter, is

the main pathway for $CO_2$ release from soil carbon pools to the atmosphere. Currently, insights into HR rates and controls are mostly derived from local-scale observations. Ecosystem respiration, or the combination of autotrophic and heterotrophic respiration fluxes, can be isolated from eddy covariance net ecosystem exchange observations at spatial scales around $1 \, km^2$, but with substantial uncertainty (Baldocchi, 2008; Barba et al., 2018; Lavigne et al., 1997). The bulk of ecosystem respiration fluxes come from soils, but soil respiration fluxes from chamber measurements can exceed ecosystem respiration measurements from flux towers, highlighting uncertainties in integrating spatial and temporal variability in ecosystem and soil respiration measurements (Barba et al., 2018). Further partitioning of soil respiration measurements into autotrophic and heterotrophic components to derive their appropriate environmental sensitivities remains challenging but critical to determining net ecosystem exchange of $CO_2$ with the atmosphere (Bond-Lamberty et al., 2004, 2011, 2018). Additionally, because fine-scale variations in environmental drivers such as soil type and soil moisture affect rates of soil respiration, it is difficult to scale local respiration observations to regional or global levels (Zhao et al., 2017). Currently, insights into HR rates and controls are mostly derived from local-scale observations. Soil chamber observations can be used to measure soil respiration at spatial scales on the order of $100 \, cm^2$ (Davidson et al., 2002; Pumpanen et al., 2004; Ryan and Law, 2005).

Local-scale observations reveal that HR is sensitive to numerous climate drivers, including temperature, moisture, and freeze–thaw state (Baldocchi, 2008; Barba et al., 2018; Lavigne et al., 1997). Because of these links to climate, predicting the evolution of HR and soil carbon stocks within coupled Earth system models is necessary for climate predictions. Within prognostic models, heterotrophic respiration has been represented as a first-order decay process based on precipitation, temperature, and a linear relationship with available substrate (Jenkinson et al., 1990; Parton, 1996; Randerson et al., 1996). However, such representations may neglect key processes for the formation of soil and persistence of soil organic carbon (SOC) stocks (Lehmann and Kleber, 2015; Schmidt et al., 2011; Rasmussen et al., 2018). More recently, models have begun to explicitly represent microbial processes in global-scale simulations of the formation and turnover of litter and SOC (Sulman et al., 2014; Wieder et al., 2013) as well as to evaluate microbial trait-based signatures on SOC dynamics (Wieder et al., 2015). These advances in the representation of SOC formation and turnover increase capacities to test emerging ideas about soil C persistence and vulnerabilities, but they also increase the uncertainties in how to implement and parameterize these theories in models (Bradford et al., 2016; Sulman et al., 2018; Wieder et al., 2018).

Given these uncertainties, developing methods to benchmark model representations of HR fluxes is an important research goal (Bond-Lamberty et al., 2018) as model predictions for soil carbon changes over the 21st century are highly uncertain (Schuur et al., 2018; Todd-Brown et al., 2014). A common method for model evaluation is to directly compare spatial or temporal variations in model properties (e.g., leaf area index) or processes (e.g., gross primary productivity) against observations (Randerson et al., 2009; Turner et al., 2006). Such comparisons assess model fidelity under present-day climate, but they may not ensure future predictability of the model. The use of functional response metrics, which evaluate the relationship between a model process and an underlying driver, may ensure that the model captures the sensitivities required to predict future evolution (Collier et al., 2018; Keppel-Aleks et al., 2018). A third benchmarking approach is to use hypothesis-driven approaches or experimental manipulations to evaluate processes (Medlyn et al., 2015). It is likely that these methods will have maximum utility when combined within a benchmarking framework (e.g., Collier et al., 2018; Hoffman et al., 2017) since they evaluate different aspects of model predictive capability.

Although a lack of direct respiration observations remains a gap for model evaluation, indirect proxies for respiration may be obtained from atmospheric $CO_2$, which reflects the balance of all carbon exchange processes between the atmosphere and biosphere. Previous work has shown that atmospheric $CO_2$ observations are inherently sensitive to HR across a range of timescales. For example, at seasonal timescales, improving the parameterization for litterfall in the CASA model improved phasing – i.e., the timing of seasonal maxima, minima, and inflection points – for the simulated annual atmospheric $CO_2$ cycle (Randerson et al., 1996). At interannual timescales, variations in the Northern Hemisphere $CO_2$ seasonal minimum are hypothesized to arise from variations in respiration (Wunch et al., 2013), and variations in the growth rate have been linked to tropical respiration and its temperature sensitivity (Anderegg et al., 2015). Here, we hypothesize that atmospheric $CO_2$ data can be used to evaluate simulations of soil heterotrophic respiration and differentiate between the chemical and microbial parameterizations used in state-of-the-art models. In this analysis, we simulate atmospheric $CO_2$ distributions using three different soil model representations that are part of a soil biogeochemical test bed (Wieder et al., 2018). The three sets of HR fluxes, shown by Wieder et al. (2018) to have distinct patterns at seasonal timescales, are used as boundary conditions for a three-dimensional atmospheric transport model. We evaluate temporal variability in the resulting $CO_2$ simulations against observations, quantify the functional relationships between $CO_2$ variability and temperature variability, and quantify the regional influences of land carbon fluxes on global $CO_2$ variability. The methods and results are presented in Sects. 2 and 3, and discussion of the implications for benchmarking and our understanding of drivers of atmospheric $CO_2$ variability are presented in Sect. 4.

## 2 Data and methods

We used a combined biosphere–atmosphere modeling approach to diagnose the signatures of land fluxes on atmospheric $CO_2$ (Fig. 1). At the heart of this approach is comparison of simulated atmospheric $CO_2$ owing to individual processes and regions to atmospheric $CO_2$ observations. The observations and models used are described below.

### 2.1 Observations and time series analysis

For this analysis we use reference $CO_2$ measurements reported in parts per million (ppm) from 34 marine boundary layer (MBL) sites (Table S1 in the Supplement) within the NOAA Earth System Research Laboratory sampling network (ESRL, Fig. 2; Dlugokencky et al., 2016). These sites were chosen to minimize the influence of local anthropogenic emissions and had at least 50 % data coverage over the 29-year period between 1982 and 2010. Following the approach in Keppel-Aleks et al. (2018), we aggregate site-specific $CO_2$ by averaging measurement time series across six latitude zones (Fig. 2, solid lines): Northern Hemisphere high latitudes (NHL: 61 to 90° N), midlatitudes (NML: 24 to 60° N), and tropics (NT: 1 to 23° N); Southern Hemisphere tropics (ST: 0 to 23° S); and two southern extratropics bands: the southern midlatitudes (SML, 24–60° S) and the southern high latitudes (SHL, 61–90° S). The global-mean $CO_2$ time series is constructed as an area-weighted average of these six atmospheric zones.

We detrend all time series data using a third-order polynomial fit to remove the impact of annually increasing atmospheric concentration in our seasonal and interannual calculations (Fig. S1 in the Supplement). Using the detrended $CO_2$ data, we calculate a period median annual cycle by averaging all observations for a given calendar month. To calculate $CO_2$ interannual variability ($CO_2$ IAV), the median annual cycle is subtracted from the detrended time series (Fig. S1). The magnitude of $CO_2$ IAV is calculated as 1 standard deviation on the detrended, deseasonalized time series, unless otherwise noted. Model-simulated $CO_2$ seasonality and interannual variability is calculated using the same methods.

### 2.2 Soil test-bed representations of heterotrophic respiration

We used a soil biogeochemical test bed (Fig. 1; Wieder et al., 2018), which generates daily estimates of soil carbon stocks and fluxes at global scale without the computational burden of running a full land model. All test-bed fluxes are output in grams of carbon per meter square ($gC\,m^{-2}$) at a daily temporal resolution and then converted to petagrams (PgC) over a region. The test bed is a chain of model simulations where soil models with different structures can be run under the same forcing data, including the same gross primary productivity (GPP) fluxes, soil temperature, and soil moisture.

The test bed produces its own estimates of net primary production (NPP), the difference between GPP and autotrophic respiration (AR; Eq. 1). Each test-bed soil model in this analysis produces unique gridded heterotrophic respiration (HR) values based on its own underlying mechanism and soil C stocks. Currently, the test bed is run with a carbon-only configuration.

For the simulations described in this paper, the modeling chain starts with the Community Land Model 4.5 (CLM4.5; Oleson et al., 2013), run with satellite phenology with CRUNCEP climate reanalysis as forcing data (Jones et al., 2012; Kalnay et al., 1996; Le Quéré et al., 2018). In this simplified formulation of CLM, a single plant functional type is assumed in each 2° by 2° grid cell. Daily values for gross primary productivity (GPP), soil moisture, soil temperature, and air temperature from CLM4.5 are passed to the Carnegie–Ames–Stanford Approach terrestrial model (CASA-CNP; Potter et al., 1993; Randerson et al., 1996, 1997; Wang et al., 2010). The CASA-CNP model uses the data from CLM4.5 to calculate NPP and carbon allocation to roots, wood, and leaves. This module also determines the timing of litterfall. Finally, metabolic litter, structural litter, and decomposing coarse woody debris (CWD) are then passed to the soil biogeochemical models to simulate HR.

From the test-bed output we calculate the net ecosystem productivity (NEP; Eq. 3). In the analysis presented here, CASA NPP was used across the test-bed ensemble in the NEP calculation, thus highlighting differences in the timing and magnitude of HR fluxes from the individual soil models. From a land perspective (positive NEP fluxes into land), NEP is calculated as NPP – HR, where respiration release of $CO_2$ decreases net carbon gains through photosynthesis. Here, we use an atmospheric perspective for NEP (positive NEP fluxes into the atmosphere) by reversing the sign on the NPP flux and taking HR as positive (Eq. 3).

$$NPP = GPP - AR \qquad (1)$$
$$NEP = HR + (-NPP) \qquad (2)$$

The three soil models make distinct assumptions about microbial processes. More details regarding these formulations and their implementation in the test bed are found in Wieder et al. (2018), but we provide brief descriptions here. The CASA-CNP soil model computes first-order, linear decay rates modified by soil temperature and moisture, implicitly representing microbial activity and soil carbon turnover through a cascade of organic matter pools (CASA: Randerson et al., 1997; CASA-CNP: CASA carbon cycling with additional nitrogen, and phosphorus cycling, Wang et al., 2010). These include metabolic and structural litter, as well as fast, slow, and passive soil carbon pools. The Microbial-Mineral Carbon Stabilization model (MIMICS; Wieder et al., 2014, 2015) explicitly represents microbial activity with a temperature-sensitive reverse Michaelis–Menten kinetics (Buchkowski et al., 2017; Moorhead and Weintraub, 2018)

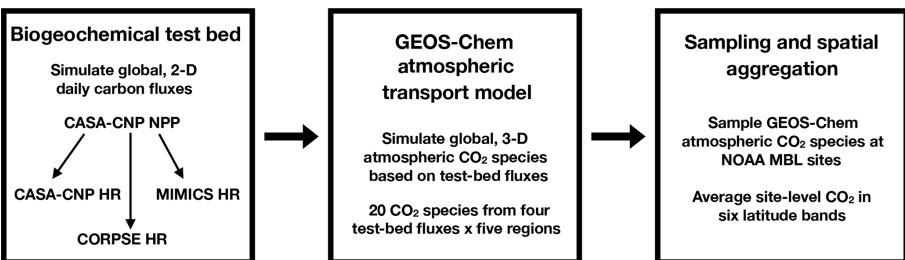

**Figure 1.** Flow chart depiction of the analysis process from soil model fluxes to simulated CO$_2$ concentration and comparison with NOAA observations.

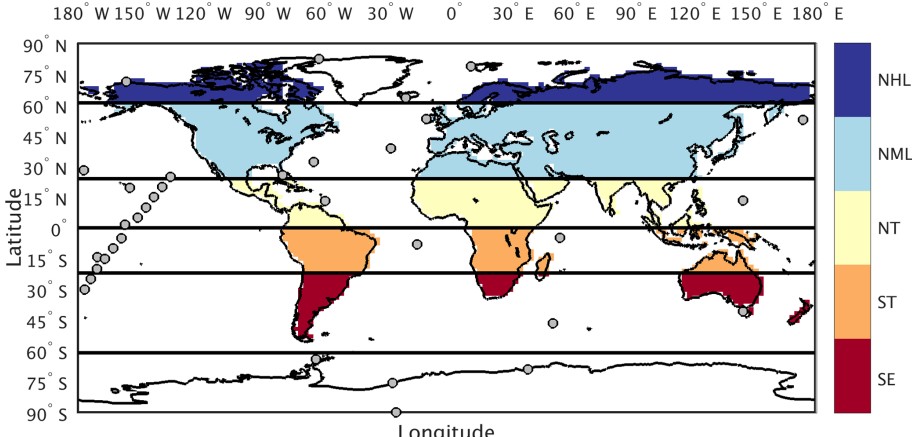

**Figure 2.** Tagged flux regions and marine boundary layer CO$_2$ observing sites used in our analysis. The five tagged flux regions are shown in color fill: northern high latitudes (NHL), northern midlatitudes (NML), northern tropics (NT), southern tropics (ST), and southern extratropics (SE). For sampling simulated CO$_2$ consistent with the tagged flux regions, we aggregate marine boundary layer sites (filled circles) into six latitude bands defined by the black lines.

but has no soil moisture controls. The decomposition pathway is set up with two litter pools (identical to those simulated by CASA-CNP), three soil organic matter pools (available, chemically and physically protected), and two microbial biomass pools for copiotrophic (fast) and oligotrophic (slow) microbial functional groups. The Carbon, Organisms, Rhizosphere, and Protection in the Soil Environment (CORPSE) model is also microbially explicit and uses reverse Michaelis–Menten kinetics, but it assumes different microbial and soil carbon pools. Surface litter and soil C pools are considered separately, but only soil C has a parallel set of physically protected pools that are isolated from microbial decomposition. CORPSE includes a temperature-dependent maximum reaction velocity ($V_{\mathrm{max}}$) parameter, but it also includes a term for the soil moisture controls on decomposition rates that uses volumetric liquid soil water content. For all three models, soil texture inputs were also derived from the CLM surface dataset (Oleson et al., 2013). We acknowledge that one potential limitation of the approach is a lack of vertical resolution in terms of temperature or frozen fraction of soil moisture (Koven et al., 2013). Overall, while the test-bed approach contains necessary simplifications, it

provides the ability to query the role of model structure, including assumptions about the number of soil carbon pools, the role of microorganisms, and the sensitivity to environmental factors, in driving HR flux differences when NPP and environmental controls are held in common.

The test-bed fluxes are used in two ways: first, we analyze monthly-averaged, regional fluxes for net primary production (NPP) from CASA-CNP and HR simulated by CASA-CNP, CORPSE, and MIMICS. Second, we use the raw daily fluxes as boundary conditions for global GEOS-Chem runs to simulate the influence of these fluxes on atmospheric CO$_2$, as described in the following section.

## 2.3  GEOS-Chem atmospheric transport modeling of CO$_2$

We simulate the imprint of the test-bed fluxes on atmospheric CO$_2$ using GEOS-Chem, a 3-D atmospheric transport model. We run the GEOS-Chem v12.0.0 CO$_2$ simulation between 1980 and 2010 at a resolution of 2.0° in latitude by 2.5° in longitude with 47 vertical levels. The model is driven by hourly meteorological data from the Modern-Era Retrospective analysis for Research

and Applications version 2 (MERRA-2) reanalysis data (Gelaro et al., 2017; http://geoschemdata.computecanada.ca/ExtData/GEOS_2x2.5/MERRA2/, last access: January 2019), with the dynamic time step set to be 600 s. The model is initialized with globally uniform atmospheric CO$_2$ mole fraction equal to 350 ppm. The test-bed fluxes from 1980 to 2010 are used for land emissions to simulate the imprint of these different soil model configurations on atmospheric CO$_2$ (Fig. 1). In our simulations, HR and NPP fluxes were separated into the five regions listed above (NHL, NML, NT, ST, SE) so that the influence of carbon fluxes originating from these individual regions on global atmospheric CO$_2$ mole fraction could be quantified. We initialized separate species of CO$_2$ in the atmospheric model, one for each flux (HR or NPP) and region (NHL, NML, etc.). Since we considered four fluxes $\boxed{\text{TS1}}$(CASA NPP and three types of HR) originating in five regions, we simulated a total of 20 species. These species were tracked throughout the simulation as their spatiotemporal distribution changed due to the combined influence of CO$_2$ fluxes at the surface and atmospheric weather. Although these species are simulated individually, we can simply sum the regional atmospheric species for a given flux (e.g., CASA HR) to determine the atmospheric CO$_2$ arising from all fluxes over the globe. We also simulated the fossil and ocean imprint on atmospheric CO$_2$ using boundary conditions from CO$_2$ CAMS inversion 17r1 (https://atmosphere.copernicus.eu/sites/default/files/2018-10/CAMS73_2015SC3_D73.1.4.2-1979-2017-v1_201807_v1-1.pdf, last access: May 2019). However, at the temporal scales of this analysis, ocean and fossil fuel fluxes had a much smaller influence on regional patterns of atmospheric CO$_2$ than did land fluxes. Across the six latitude bands, the detrended CO$_2^{\mathrm{NEP}}$ annual amplitude ranges from a factor of 1.5 (in the tropics) to an order of magnitude larger (at high latitudes) than CO$_2$ from ocean fluxes and fossil fuel emissions. Likewise, the IAV from fossil and ocean-derived CO$_2$ was at most 25 % that of NEP-derived CO$_2$ at most latitude bands. These results are consistent with previous studies that have demonstrated that NEP drives most of the atmospheric CO$_2$ seasonality ($>$90 %; Nevison et al., 2008; Randerson et al., 1997) and interannual variability (e.g., Rayner et al., 2008; Battle et al., 2000). Given that patterns of IAV in ocean and fossil CO$_2$ partially cancel each other and the large uncertainty in ocean fluxes, we choose to omit these CO$_2$ species from our analysis.

We discard the first 2 years of the atmospheric simulations for model spin-up, and we analyze the monthly average model outputs for the period 1982–2010. We sample the gridded atmospheric simulation output at the 34 marine boundary layer (MBL) sites identified in Sect. 2.1, using the third vertical level to minimize influence of land–atmosphere boundary layer dynamics. We then calculate the latitude zone average, median annual cycle and interannual variability using the methods described for CO$_2$ observations (see Sect. 2.1). Averaging CO$_2$ from all sites within a latitude band is consis-

tent with our hypothesis that atmospheric CO$_2$ may provide constraints on large-scale patterns of heterotrophic respiration, but individual sites may be too heavily influenced by local characteristics not accounted for by the test-bed fluxes. As such, averaging simulated and observed CO$_2$ across latitude zones smooths local information while retaining information about regional-scale fluxes.

Throughout the paper, we refer to CO$_2$ originating from these NPP and HR component fluxes as CO$_2^{\mathrm{NPP}}$ and CO$_2^{\mathrm{HR}}$, respectively. We use a sign convention for the fluxes whereby a positive value indicates a source of carbon to the atmosphere, which means we can combine the CO$_2$ tracers from NPP and HR to calculate the expected atmospheric variation owing to NEP using (Eq. 3):

$$CO_2^{\mathrm{NEP}} = CO_2^{\mathrm{HR}} + CO_2^{\mathrm{NPP}}. \tag{3}$$

We note that the net CO$_2$ response from the model (i.e., CO$_2^{\mathrm{NEP}}$) is approximately equivalent to observations in terms of seasonal and interannual variations, although we neglect ocean fluxes and emissions from fossil fuels, land use and land cover change, and disturbance. In the results below, the superscript notation will be used to denote the test-bed ensemble sources. For example, CO$_2^{\mathrm{HR}}$ simulated from CORPSE fluxes is defined as CO$_2^{\mathrm{CORPSE\ HR}}$, similarly for CO$_2^{\mathrm{CORPSE\ NEP}}$.

## 2.4 Global temperature sensitivity and separation of regional influences

For insight into a functional climate response, we investigate the global temperature sensitivity of the atmospheric CO$_2$ growth rate and the test-bed ensemble fluxes. Rates of change were derived from monthly and annual time series to calculate the temperature sensitivity of the test-bed fluxes, the modeled CO$_2$, and the observed CO$_2$ values. The CO$_2$ growth rate anomaly was calculated as the difference between time step $n$ and $n - 1$ in both the monthly and annual CO$_2$ IAV time series. As a result of this technique, the monthly CO$_2$ growth rate anomalies were centered on the first day of the corresponding months. To compare flux information with CO$_2$ growth rate anomalies, daily test-bed flux time series were averaged to monthly resolution and then interpolated by averaging between months to center values on the first day of each month.

Following Arora et al. (2013), we calculate temperature sensitivity ($\gamma$) using an ordinary linear regression (OLR). We calculate OLR for the interannual variability time series of CASA-CNP soil temperature (T IAV) against (1) atmospheric CO$_2$ growth rate anomalies and (2) land flux IAV (see Sect. 2.2). For atmospheric CO$_2$ growth rate anomalies, each time series was converted from parts per million per year to petagrams of carbon per year based on the global mass of atmospheric dry air. Thus, all global temperature sensitivity values are reported in units of petagrams of carbon per year per kelvin. The global temperature sensitivity value for

the observed $CO_2$ growth rate anomaly was calculated for 1982 to 2010 using ESRL $CO_2$ observations and the Climatic Research Unit's gridded temperature product (TS2 CRU TS4; Jones et al., 2012), which is derived from interpolated ground station measurements.

We also assess the influence of individual regions on the global-mean signal for both component land fluxes (NPP, HR) and simulated atmospheric $CO_2$ ($CO_2^{NPP}$, $CO_2^{HR}$, $CO_2^{NEP}$). We first quantify the magnitude of variability in each region relative to the magnitude of global variability ($\sigma_{REL}$) as the ratio of regional IAV standard deviation to global IAV standard deviation. This ratio is calculated for monthly flux IAV from each of the five flux regions and for the global-mean $CO_2$ time series that arises from fluxes in each of the five flux regions (e.g., the global $CO_2$ response to NHL fluxes, or the global $CO_2$ response to NML fluxes). The value of $\sigma_{REL}$ has a lower bound of 0, which would indicate that a region contributes no IAV, but has no upper bound, since a value greater than 1 simply indicates that the fluxes in a given region are more variable than global fluxes.

We note that the timing of IAV in a given region may be independent of IAV in other regions and thus may or may not be temporally in-phase with global IAV. We therefore also calculate correlation coefficients ($r$) for the time series of regional flux IAV and $CO_2$ IAV with the global signal. Thus, if an individual region were responsible for all observed global flux or $CO_2$ variability, it would have both $\sigma_{REL}$ and $r$ values equal to 1 in this comparison. The value for $r$ will be small if a regional signal is not temporally coherent with the global signal, even if the magnitude of variability is high.

## 3 Results

### 3.1 Seasonal imprint of heterotrophic respiration

Our evaluation of $CO_2$ simulated using test-bed fluxes revealed that all test-bed models overestimated the mean annual cycle amplitude of atmospheric $CO_2$ observations. In the Northern Hemisphere, the bias was largest for MIMICS, as the $CO_2^{MIMICS\ NEP}$ amplitude was overestimated by up to 100 % (Fig. 3). The mismatch was smallest in $CO_2^{CORPSE\ NEP}$, which was within 70 % of the observed annual cycle amplitude where CORPSE simulates the largest seasonal HR fluxes (Fig. 3a–c, Table 1). Within the modeled carbon dioxide concentrations resulting from land fluxes, $CO_2^{NPP}$ and $CO_2^{HR}$ show the largest seasonality in the NHL, with seasonal amplitudes decaying toward the tropics and Southern Hemisphere. In the NHL, the peak-to-trough amplitude of $CO_2^{NPP}$ is $39 \pm 2$ ppm, with a seasonal maximum in April and a seasonal minimum in August (Fig. 4a; note this $CO_2^{NPP}$ peak reflects the sign reversal in the driving NPP flux (Sect. 2.3)). The seasonal cycles for $CO_2^{HR}$ simulated from all test-bed models are out of phase with that of $CO_2^{NPP}$, and there are large amplitude differences in

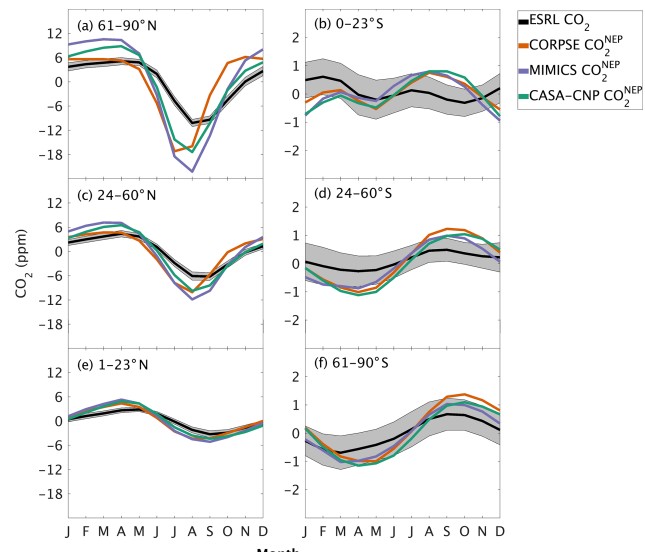

**Figure 3.** Climatological annual cycle (median) of $CO_2$ for observations (black) and global net ecosystem productivity flux ($CO_2^{NEP}$, colors) between 1982 and 2010. Monthly climatology values were created after detrending the $CO_2$ time series for atmospheric sampling bands in the **(a–c)** Northern Hemisphere **(d–f)** and Southern Hemisphere. Note the change in $y$-axis scale between the two hemispheres and the sign of $CO_2^{NEP}$ reflects the combination of $CO_2^{NPP}$ and $CO_2^{HR}$ (Eq. 3). Shading on the observed line represents one standard deviation due to interannual variability in the seasonal cycle.

$CO_2^{HR}$ among the model ensemble members. Specifically, the NHL amplitude of $CO_2^{CORPSE\ HR}$ is $28 \pm 3$ ppm, while the amplitudes for $CO_2^{MIMICS\ HR}$ and $CO_2^{CASA-CNP\ HR}$ are only $17 \pm 1$ ppm, accounting for about 40 %–70 % of the amplitude from $CO_2^{NPP}$ (Table 1). However, in all latitude bands, the largest $CO_2^{HR}$ amplitude comes from the microbially explicit model – CORPSE for the Northern Hemisphere. In the Southern Hemisphere extratropics, the amplitudes for all components were less than 3 ppm (Table 1).

The three soil carbon models in the test bed impart different fingerprints on atmospheric $CO_2$ variability. Specifically, the phasing of $CO_2^{HR}$ is an important driver of the overall comparison between $CO_2^{NEP}$ and observed $CO_2$ seasonality (Fig. 3). When the contributions of NPP and HR seasonality are considered together (i.e., $CO_2^{HR} + CO_2^{NPP}$), the simulated amplitude of $CO_2^{NEP}$ is larger than the observed $CO_2$ across all latitude bands (Fig. 3). The largest mismatch is in the NHL zone, where the observed mean annual cycle is $15 \pm 0.9$ ppm, while the peak-to-trough $CO_2^{NEP}$ ranges from $23 \pm 1.3$ ppm for CORPSE to $33 \pm 1.4$ ppm for MIMICS (Fig. 3a). The smaller $CO_2^{NEP}$ amplitude simulated by CORPSE is due to the large $CO_2^{HR}$ seasonality that counteracts the seasonality in NPP (Fig. 4a–b). Furthermore, $CO_2^{MIMICS\ HR}$ and $CO_2^{CASA-CNP\ HR}$ have similar amplitudes in the NHL (Fig. 4a; Table 1), but the $CO_2^{NEP}$ amplitude

**Table 1.** Atmospheric $CO_2$ mean annual cycle amplitude (in ppm) simulated from heterotrophic respiration (HR), net primary productivity (NPP), and net ecosystem productivity (NEP). The median annual cycle amplitudes for observed $CO_2$ ($CO_2^{OBS}$) averaged over latitude bands are also reported.

|  | 61–90° N | 24–60° N | 0–23° N | 1–23° S | 24–60° S | 61–90° S |
|---|---|---|---|---|---|---|
| $CO_2^{\text{CASA-CNP HR}}$ | 17.6 | 11.4 | 4.3 | 4.3 | 1.1 | 1.9 |
| $CO_2^{\text{CORPSE HR}}$ | 28.2 | 16.6 | 6.4 | 4.9 | 1.4 | 2.2 |
| $CO_2^{\text{MIMICS HR}}$ | 17.2 | 11.8 | 5.1 | 4.4 | 1.9 | 2.5 |
| $CO_2^{\text{CASA-CNP NPP}}$ | 39.3 | 24.6 | 11.9 | 6.0 | 3.1 | 3.1 |
| $CO_2^{\text{CASA-CNP NEP}}$ | 26.2 | 16.3 | 9.3 | 1.6 | 2.2 | 2.2 |
| $CO_2^{\text{CORPSE NEP}}$ | 23.4 | 14.8 | 8.7 | 1.3 | 2.2 | 2.4 |
| $CO_2^{\text{MIMICS NEP}}$ | 32.8 | 19.0 | 10.4 | 1.7 | 1.9 | 2.1 |
| $CO_2^{\text{OBS}}$ | 15.3 | 10.6 | 6.1 | 0.9 | 0.8 | 1.4 |

from these two models differs ($33 \pm 1.2$ ppm versus $26 \pm 1$ ppm, respectively; Fig. 3a; Table 1). This occurs because $CO_2^{\text{MIMICS HR}}$ peaks 1 month later than $CO_2^{\text{CASA-CNP HR}}$ and has a zero crossing that is more closely aligned with the trough of $CO_2^{NPP}$ (Fig. 4a), leading to the larger amplitude in $CO_2^{\text{MIMICS NEP}}$ (Fig. 3a; Table 1). Although the amplitude mismatch decreases towards the south (Fig. 3b–f), the overall bias in the Northern Hemisphere suggests that either the seasonality of NPP is too large or that all test-bed models underestimate the seasonality of HR. Within the ST region, ensemble $CO_2^{HR}$ minima are opposite to those in $CO_2^{NPP}$, leading to a small annual cycle in simulations, consistent in magnitude with that of the observations (Figs. 3d, 4d).

### 3.2 Interannual imprint of heterotrophic respiration

The test-bed ensemble reasonably simulates the magnitude and timing of interannual variability (IAV) compared with $CO_2$ observations (Fig. 5). Across the six latitude bands analyzed, simulated $CO_2^{NEP}$ IAV generally falls within 1 standard deviation of the median variation from observations for most of the study period (Fig. 5). Taking a closer look at the $CO_2$ from the component fluxes (NPP and HR), across all six latitude bands, the $CO_2^{NPP}$ IAV standard deviation is between 0.9 and 1.1 ppm (Fig. 6b). $CO_2^{\text{CASA-CNP HR}}$ IAV shows standard deviation similar to that of $CO_2^{NPP}$ IAV, whereas the standard deviations of $CO_2^{\text{CORPSE HR}}$ and $CO_2^{\text{MIMICS HR}}$ range from 0.7 to 1.4 ppm and 0.5 to 1.1 ppm, respectively (Fig. 6b).

Combining the $CO_2$ responses from component fluxes to $CO_2^{NEP}$ reveals a latitudinal gradient in IAV standard deviation similar to that of ESRL observations, with the largest standard deviation found in the northern extwhere

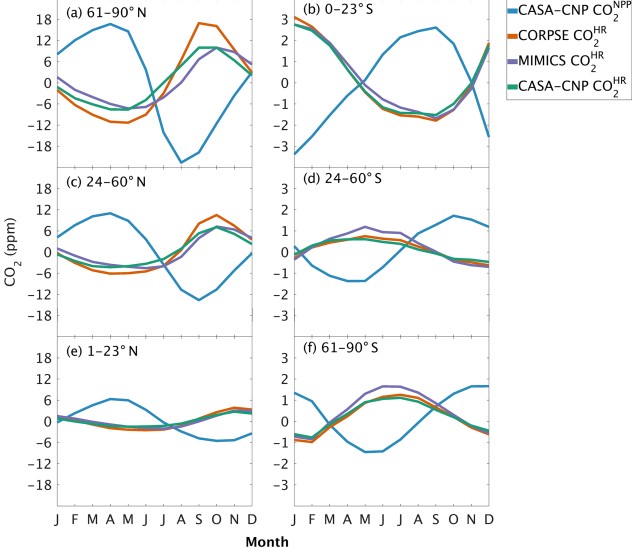

**Figure 4.** Climatological annual cycle (median) of atmospheric $CO_2$ simulated from land fluxes ($CO_2^{NPP}$, $CO_2^{HR}$) between 1982 and 2010. Monthly climatology values were created after detrending the $CO_2$ time series for atmospheric sampling bands in the **(a–c)** Northern Hemisphere **(d–f)** Southern Hemisphere. Note the change in $y$-axis scale between the two hemispheres, and the sign of $CO_2^{NPP}$ reflects the sign reversal of the underlying NPP (positive flux to the atmosphere; Eq. 2).

ics (Fig. 6a). Among the three test-bed models, the standard deviation of $CO_2^{\text{CASA NEP}}$ agrees best with observations across all latitude bands ($CO_2^{\text{CASA NEP}}$: 0.5–0.9 ppm; ESRL: 0.6–1.0 ppm; Fig. 6a). $CO_2^{\text{CORPSE NEP}}$ overestimates IAV by up to 30 % in NHL and NML but agrees better with observa-

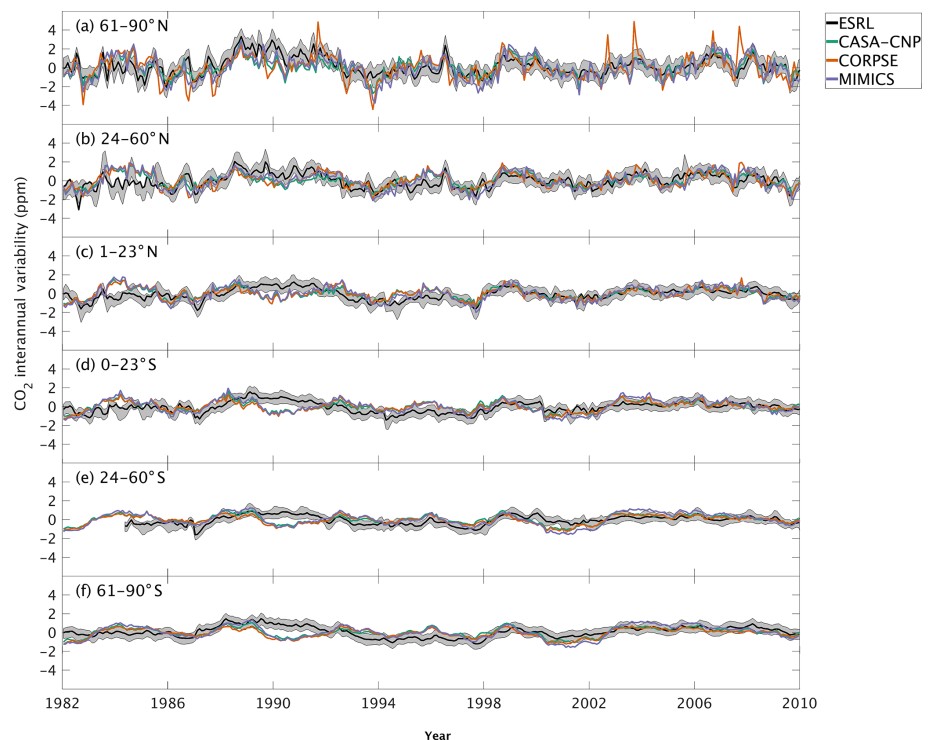

**Figure 5.** Interannual variability of $CO_2$ from global net ecosystem productivity ($CO_2^{NEP}$ IAV) for test-bed models (colors) and marine boundary layer observations from the NOAA ESRL network (black). Gray shading outlines 1 standard deviation of observed $CO_2$ interannual variability. High-latitude, midlatitude, and tropical land belts are shown for the Northern Hemisphere **(a–c)** and Southern Hemisphere **(d–f)**.

tions in the tropics and Southern Hemisphere. $CO_2^{MIMICS\ NEP}$ overestimates IAV standard deviations across all latitude bands (Fig. 6a). Interestingly, in the NHL, the overestimation is 20 % even though $CO_2^{MIMICS\ HR}$ shows IAV similar to that of $CO_2^{NPP}$ (both 1.1 ppm; Fig. 6b). This suggests that the atmospheric $CO_2$ diagnostic for IAV, like that for amplitude, is critically sensitive to the phasing of IAV in heterotrophic respiration relative to the IAV of NPP.

Both global NPP and HR fluxes are sensitive to temperature variations at interannual timescales, with increased buildup of $CO_2$ in the atmosphere at higher temperatures, in part because the rate of HR increases at higher temperature and in part because most latitude bands show a reduction in NPP at above-average temperatures. For CASA-CNP, the temperature sensitivity ($\gamma$) for globally integrated NPP and HR fluxes is 2.5 and 1.7 PgC yr$^{-1}$ K$^{-1}$, respectively (Fig. 7b). The temperature sensitivity of HR was higher for the microbially explicit models: 2.1 PgC yr$^{-1}$ K$^{-1}$ for CORPSE and 4.2 PgC yr$^{-1}$ K$^{-1}$ for MIMICS (Fig. 7b). For any given test-bed flux (NPP, HR, or NEP), the temperature sensitivity of the resulting global-mean $CO_2$ growth rate anomaly is higher than that of the underlying flux IAV. For example, the temperature sensitivity of the globally integrated NPP flux IAV ($\gamma$NPP) is 2.5 PgC yr$^{-1}$ K$^{-1}$ whereas $\gamma CO_2^{NPP}$ is 3.2 PgC yr$^{-1}$ K$^{-1}$. The apparent amplification of the temperature sensitivity was even larger

for HR. For example, the temperature sensitivity of MIMICS HR IAV ($\gamma$HR$^{MIMICS}$) was 4.2 PgC yr$^{-1}$ K$^{-1}$, whereas $\gamma CO_2^{MIMICS\ HR}$ was 7.7 PgC yr$^{-1}$ K$^{-1}$ (Fig. 7b). The simulated $\gamma CO_2^{NEP}$ simulated by the test-bed models all overestimate the temperature sensitivity of the observed atmospheric $CO_2$ growth rate anomaly (6.1 ± 2.5 PgC yr$^{-1}$ K$^{-1}$; Fig. 7a). CASA-CNP and CORPSE have temperature sensitivities within the range of the observed sensitivity (5.16 ± 0.9 PgC yr$^{-1}$ K$^{-1}$, Cox et al., 2013; 6.5 ± 1.8 PgC yr$^{-1}$ K$^{-1}$; Keppel-Aleks et al., 2018), but $\gamma CO_2^{MIMICS\ NEP}$ is 80 % larger than the observed value (10.9 PgC yr$^{-1}$ K$^{-1}$; Fig. 7a). We note that the $\gamma$HR and $\gamma CO_2^{HR}$ are emergent properties that reflect both direct and indirect temperature influences, including the impact of temperature variability on NPP and litterfall (Table S3). Nevertheless, these results suggest that the direct temperature sensitivity of MIMICS HR is too high relative to observational constraints.

## 3.3 Geographic origins of $CO_2$ IAV

The interannual variability (IAV) in global NPP and HR originates from different geographic regions. The IAV in global NPP fluxes is dominated by variations within the tropics (both NT and ST regions), with a relative standard deviation $\sigma_{REL} \sim 0.5$ and correlation coefficient $r \sim 0.6$ (Fig. 8a–b). The NML region also has a similar contribution to the

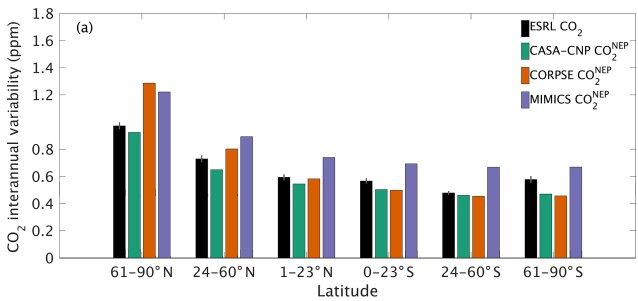

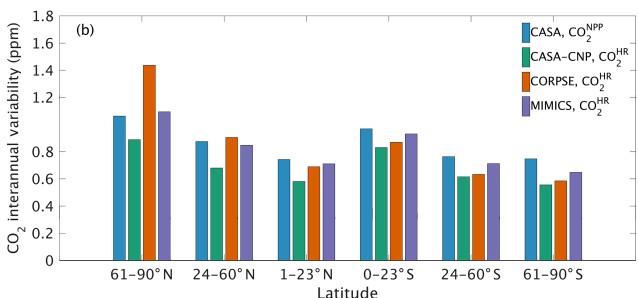

Figure 6. Magnitude of $CO_2$ interannual variability resulting from (a) $CO_2^{NEP}$ and (b) component fluxes CE1. Observed $CO_2$ IAV from the NOAA ESRL network is shown with black bars whereas colors represent simulated data. Error bars shown on the observed IAV represent 2 standard deviations, calculated as the median magnitude after removing a 12-month sliding window from the IAV time series.

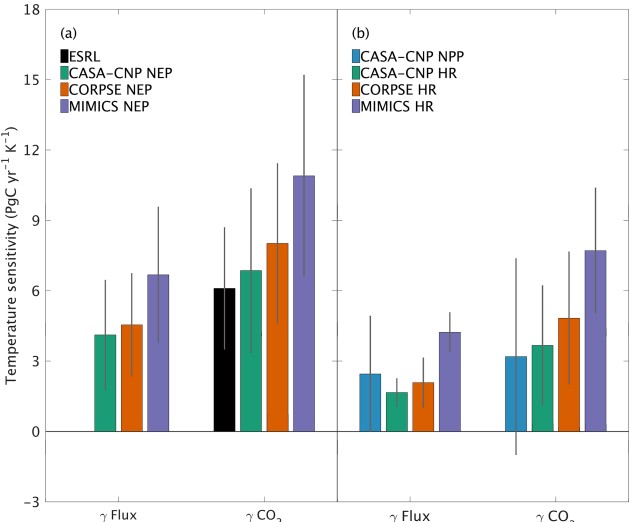

Figure 7. Temperature sensitivity ($\gamma$) calculated for interannual variability (IAV) of CASA-CNP air temperature and (a) NEP flux IAV and corresponding $CO_2$ TS4 growth rate anomalies and (b) component flux IAV and $CO_2^{NEP}$ TS5 growth rate anomalies. The reference sensitivity value (black) was calculated using NOAA ESRL $CO_2$ and CRU TS4 air temperature. Sensitivity values were calculated as the ordinary linear regression coefficient between IAV time series for 1982 to 2010. Error bars represent the 95 % confidence interval for coefficient values.

NT in magnitude, but with a lower timing coherence ($r = 0.44$; Fig. 8a–b). In contrast to the dominance of the tropics in contributing to the interannual variability of global NPP, the NML region contributes most to IAV in global HR, with $\sigma_{REL} \geq 0.6$ and $r \sim 0.8$ for all three test-bed models (Fig. 8c–d). The NHL region is also important in driving global HR flux variability based on CORPSE model results ($\sigma_{REL} = 0.59$ and $r = 0.82$; Fig. 8c–d). Despite high NPP variability in the tropics, the magnitude of tropical HR variability is only about 10 %–30 % of global HR variability, and the timing coherence with the global signal is generally low ($r < 0.45$; Fig. 8a–b). MIMICS HR IAV is the exception for the ST, measuring close to 40 % of global HR IAV magnitude and relatively high correlation ($r = 0.58$; Fig. 8c–d). Together, the tropics and NML contribute roughly equally to the magnitude of global NEP variability ($\sigma_{REL}$ between 0.44 and 0.55; Fig. 8e). Although the NML and NT show relatively high timing coherence (0.41–0.55), the ST show the strongest timing coherence with global NEP IAV ($r > 0.7$; Fig. 8f).

Atmospheric transport modifies patterns of IAV in fluxes, emphasizing tropical flux patterns and de-emphasizing Northern Hemisphere flux patterns. For example, the role of ST in driving global $CO_2^{NPP}$ variability is amplified compared to the underlying fluxes, as the timing coherence with the global signal increases from $r = 0.64$ for flux IAV to $r = 0.88$ for $CO_2^{NPP}$ IAV for this region (Fig. 8b). Conversely, the role of NML is dampened, with timing coherence decreasing to $r = 0.33$ for $CO_2^{NPP}$ IAV versus $r = 0.44$ for NPP IAV (Fig. 8b). Similarly, timing coherence for tropical $CO_2^{HR}$ IAV is substantially higher than that for HR fluxes in the ST and NT ($>0.7$), although the atmospheric transport impact differs across the three test-bed models (Fig. 8d). In contrast to closely aligned NML correlation values for $CO_2^{HR}$ and HR ($r \sim 0.8$–0.9), NML $CO_2^{HR}$ IAV shows $\sigma_{REL}$ between 0.45 and 0.58, a decrease from the HR IAV contribution (NML HR IAV $\sigma_{REL}$ range: 0.57 to 0.74; Fig. 8c). For $CO_2^{NEP}$ IAV, the regional contribution is more consistent with $\sigma_{REL}$ and $r$ similar to that of flux IAV (Fig. 8e–f). Thus, numerical effects of transport modeling should be considered when isolating the impact of regional land fluxes on global atmospheric $CO_2$.

## 4 Discussion

Modeled differences in heterotrophic respiration impart discernible signatures on atmospheric $CO_2$, suggesting that atmospheric $CO_2$ observations may be able to help evaluate broad differences in the timing and magnitude fluxes simulated by different vegetation and soil biogeochemical models. We used a 3-D atmospheric transport model to analyze the imprint of the atmospheric $CO_2$ resulting from soil heterotrophic respiration and net ecosystem exchange

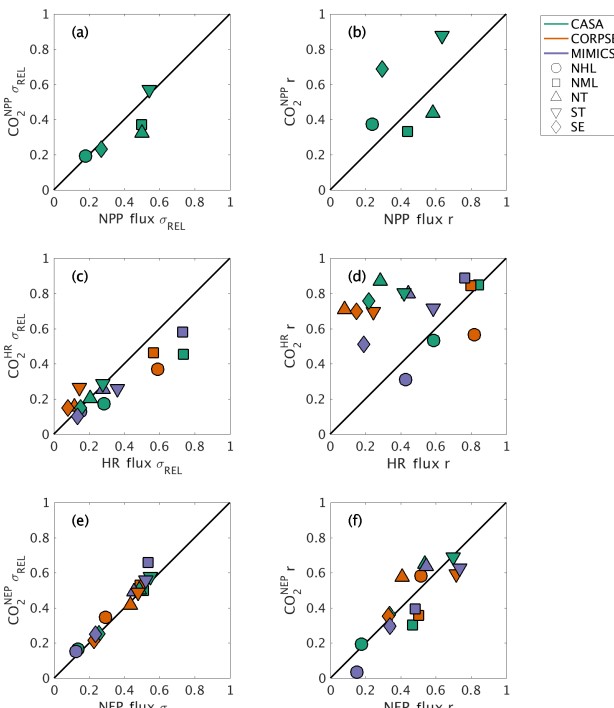

**Figure 8.** Comparison of regional and global interannual variability (IAV) from land fluxes and resulting atmospheric $CO_2$ between 1982 and 2010. **(a, c, e)** Normalized ratio taken between regional IAV and global IAV magnitude. **(b, d, f)** Linear correlation between regional IAV and global IAV. The scatterplot shows a direct comparison of ratio and correlation values for land flux values (*x* axes) and corresponding $CO_2$ (*y* axes). Shapes denote the source regions for both land fluxes and $CO_2$ response.

fluxes from the soil test-bed ensemble with three representations of soil biogeochemistry (CASA-CNP, CORPSE, MIMICS). Results show that the phasing of heterotrophic respiration fluxes relative to net productivity fluxes is an im-
5 portant source of bias in evaluating simulated $CO_2$ against atmospheric observations at both seasonal and interannual timescales. Regional patterns of heterotrophic respiration variability provide non-negligible contributions to global $CO_2$ variability. Here we discuss these findings in more de-
10 tail as well as implications for the use of $CO_2$ observations for flux evaluation and model benchmarking.

### 4.1    Impacts of heterotrophic respiration on seasonality

Our evaluation of $CO_2$ simulated using test-bed fluxes revealed that all test-bed models overestimated the mean an-
15 nual cycle amplitude of atmospheric $CO_2$ observations. In the Northern Hemisphere, the bias was largest for MIMICS, which had a $CO_2$ amplitude from net ecosystem production that was overestimated by up to 100 % (Fig. 3). The mismatch in the amplitude of the Northern Hemisphere
20 NEP fluxes was smallest from CORPSE, despite CORPSE

also simulating the largest seasonal amplitude in HR fluxes (Fig. 3a–c, Table 1). By contrast, in the Southern Hemisphere the simulated $CO_2$ annual cycle amplitudes were similar across all three models, with small absolute mismatches (about 1 ppm) compared to observations (Fig. 3). We note 25 that the differences in the amplitude of NEP fluxes across all three test-bed formulations could be due to biases in the timing and magnitude of NPP and HR fluxes simulated by models in the test bed. However, an advantage of the test-bed approach is that, because all of the models are driven by 30 the same GPP and climate variables, the differences in the timing and magnitude of NEP fluxes are all related to differences in HR fluxes that are simulated by different soil models in the test bed. With future work we would like to consider forcing uncertainty that could be generated by using different 35 inputs of productivity, temperature, and moisture from land model ensembles (e.g., TRENDY simulations, CMIP6 models). From these results, however, it appears that the seasonal amplitude of atmospheric $CO_2$ fluxes from net ecosystem production that are simulated in the northern high latitudes 40 and midlatitudes are higher than atmospheric observations for all of the models tested here, but especially MIMICS.

One challenge in using atmospheric $CO_2$ to evaluate HR representation in soil models is the influence of productivity (NPP) on both HR fluxes and atmospheric $CO_2$ varia- 45 tions. The seasonal diagnostics we present are very sensitive to the phasing of HR fluxes relative to NPP. For example, in NHL a 1-month lag in the seasonal maximum of $CO_2^{HR}$ between MIMICS and CASA-CNP (Fig. 4a) leads to a 7 ppm difference in the overall amplitude of $CO_2^{NEP}$ – this 50 despite identical amplitudes of $CO_2^{HR}$ for the two models (Fig. 3a). Although the substantial impacts of subtle phase differences complicate benchmarking, the sensitivity reveals interesting and important differences related to model structural choices (i.e., first order versus microbially explicit). 55 Wieder et al. (2018) noted that the microbially explicit models in the test bed had seasonal HR fluxes that peaked in the fall, about a month later than the HR fluxes simulated by CASA-CNP. Annual phasing of HR is altered with the addition of microbial processes but also reflects NPP seasonal- 60 ity. The timing of CASA-CNP fluxes largely depend on soil temperature (highest HR flux when temperature is highest), whereas MIMICS and CORPSE have maximum HR fluxes set by trade-offs between the timing of maximal temperature and maximal microbial biomass, which is more tightly linked 65 with litterfall (Fig. 7 from Wieder et al., 2018). Thus, phasing of HR is a sensitive diagnostic for benchmarking, especially if additional constraints on the magnitude and phasing of NPP are available.

In this study, determining the unique contribution from HR 70 was possible since NPP was common among the three soil models used in the test bed, but the contribution of NPP will need to be resolved for model evaluation in other contexts. For example, long-term records of vegetation productivity at regional and global scales have been observed via satellite 75

vegetation indices (Hicke et al., 2002; Meroni et al., 2009; Running et al., 2004) and more recently chlorophyll fluorescence (Frankenberg et al., 2011; Guan et al., 2016; Köhler et al., 2018; Li et al., 2018). Our study underscores the importance of developing methods to use these datasets together with atmospheric $CO_2$ to inform the dynamics of carbon cycling and its component fluxes. Current benchmarks used to evaluate carbon cycle metrics in land models include globally gridded estimates of fluxes (GPP, NEE, ecosystem respiration) and C stocks (leaf area index, vegetation biomass, and soil C; Collier et al., 2018). This is an excellent starting point, but it provides a rather coarse estimate for the component fluxes we are trying to evaluate with this analysis. Notably, current benchmarks but do not yet consider the other metrics like NPP, litterfall, or root turnover and exudation that are important drivers of ecosystem, soil, and heterotrophic respiration. Globally gridded estimates of annual soil respiration have been upscaled using machine learning techniques (Zhao et al., 2017), and we recognize the value in using this and similar data products to provide an independent benchmark to evaluate C fluxes that are simulated by models in the test bed or other model ensembles. These annual estimates are useful for looking at the spatial distribution of fluxes and inferring information about simulated trends, but they will not help resolve differences in the timing of heterotrophic respiration fluxes (Fig. 4) that are driving differences in net ecosystem production in the test-bed models (Fig. 3). Instead, additional work with databases of soil and heterotrophic respiration (e.g., Bond-Lamberty and Thomson, 2010; Schädel et al., 2019) will be critical to evaluating the seasonal dynamics and environmental sensitivities of soil and heterotopic respiration fluxes.

## 4.2 Impacts of heterotrophic respiration on interannual variability

Capturing appropriate interannual variability is important to generating credible land C-cycle representations (Cox et al., 2013; Piao et al., 2020). To a first approximation, all models in the test bed generated interannual variability in NEP fluxes that matched latitudinal distributions from atmospheric observations (Fig. 5). Similar to the analyses on seasonal cycles, the test-bed ensemble simulations showed a higher interannual variability of $CO_2$ fluxes associated with explicit microbial representation – especially for heterotrophic respiration fluxes with CORPSE in the northern high latitudes (Figs. 5a, 6).

Interestingly, in the tropics and southern extra-tropics, the interannual variability of heterotrophic respiration fluxes simulated by MIMICS is only slightly higher than CASA-CNP or CORPSE (Fig. 6b), but the interannual variability of NEP fluxes simulated by MIMICS was 20 %–30 % higher than that of other models (Fig. 6a). Further, in these regions the interannual variability of heterotrophic respiration fluxes simulated by MIMICS also shows an inverse but highly cor-

related relationship with the interannual variability of NPP ($R^2 > 0.60$, Table S3). This suggests that the large interannual variability of NEP fluxes simulated by MIMICS may result from differences in phasing between NPP and MIMICS HR fluxes, similar to phasing between MIMICS NPP and HR affecting the shape of the $CO_2^{NEP}$ annual cycle in northern high latitudes. In the northern high latitudes, all test-bed models show interannual variability of heterotrophic respiration is correlated with the interannual variability of both NPP and temperature ($R^2$ of 0.32 to 0.77; Table S3). Additionally, the interannual variability NPP is sensitive to temperature variability ($\gamma = 0.15$, $R^2 = 0.43$; Table S3). As in Sect. 4.1, better diagnostics to partition the interannual variability of atmospheric $CO_2$ measurements into environmental sensitivities of heterotrophic respiration and productivity are required, especially at high latitudes, but our results suggest that the carbon cycle simulated by the MIMICS model shows interannual variability of $CO_2$ fluxes that is higher than atmospheric observations.

This high interannual variability of NEP simulated by MIMICS is consistent with this model having the highest global temperature sensitivity, overestimating observed values by 80 % (Fig. 7a). CORPSE, the other microbially explicit model, had a 30 % higher temperature sensitivity in $CO_2^{NEP}$ than observed globally (Fig. 7a). This large bias in temperature sensitivity demonstrates uncertainties in the model structure and parameterization that is associated with soil biogeochemical models (Sulman et al., 2018). And although the temperature sensitivity of microbial kinetics simulated in MIMICS was parameterized with observations from enzyme assays from laboratory experiments (German et al., 2012; Wieder et al., 2014, 2015), additional factors, including substrate availability, exert important proximal controls over the ultimate temperature sensitivity of soil C decomposition (Conant et al., 2011; Dungait et al., 2012). Recently, Zhang et al. (2020) used observations from >200 sites in Europe and China to calibrate parameters for MIMICS, but these parameters have not yet been tested globally. Future work should similarly leverage local observations for model calibration to develop parameters that can be applied in subsequent global-scale simulations. The work presented here establishes a framework that uses a top-down constraint of atmospheric $CO_2$ observations to then evaluate, or benchmark, the $CO_2$ fluxes that are simulated by the revised model(s). As with larger land models (Collier et al., 2018), we see this interplay of model parameterization, testing, and evaluation as critical to refining and improving confidence in projections from soil biogeochemical models (Bradford et al., 2016).

## 4.3 Implications for model benchmarking using atmospheric $CO_2$

Our results provide useful insights for model benchmarking using atmospheric $CO_2$. On a global scale, interannual variability of simulated atmospheric $CO_2$ was shown to be

affected by the variability in component fluxes (NPP, HR) from different land regions (Figs. 6–8). The tropics dominate the interannual variability in global NPP, while northern extratropics dominate the interannual variability in global heterotrophic respiration (Fig. 8a–d). Taken together, NEP variability reflects roughly equal contributions from Northern Hemisphere temperate ecosystems (NML) and tropical ecosystems (NT and ST; Fig. 8e–f). These results suggest that the interannual variability of atmospheric $CO_2$ results from two different processes (respiration and productivity) across multiple ecoclimatological regions, whereas previous studies have mostly identified tropical (e.g., Cox et al., 2013; Wang et al., 2013) or subtropical, semiarid regions (e.g., Ahlström et al., 2015; Poulter et al., 2014) as dominant controls on the global interannual variability of atmospheric $CO_2$ observations. Additional analyses are needed to test the robustness of this finding with different forcings and soil models, but these results emphasize the importance of different processes and regions as sources of variability in the terrestrial carbon cycle.

Our analysis underscores that patterns of variability in atmospheric $CO_2$ are tied not only to variations in the underlying fluxes, but also to atmospheric transport. For example, we showed that the temperature sensitivity of $CO_2$ growth rate anomalies was larger than the sensitivity estimated from the fluxes themselves (Fig. 7). The enhanced temperature sensitivity for $CO_2^{HR}$ was larger than for that of $CO_2^{NPP}$, which suggests that the geographic origin of the fluxes relative to dominant patterns of transport affects the result (Fig. 7b). This transport enhancement of the apparent temperature sensitivity of $CO_2$ growth rate anomalies is consistent with results from Keppel-Aleks et al. (2018). While these results may be tied to the choice of GEOS-Chem to simulate atmospheric transport, they do underscore that (1) atmospheric $CO_2$ must be simulated from land fluxes to be used as a benchmark and (2) atmospheric observations should not be assumed to be a direct proxy for fluxes themselves.

We employed several benchmarking approaches, including time series comparison and functional response to temperature, to evaluate if $CO_2$ patterns reflect underlying representations of soil heterotrophic respiration. We found that soil heterotrophic respiration leaves non-negligible imprints on atmospheric $CO_2$, leaving open the possibility of more explicitly accounting for respiration variability using atmospheric $CO_2$ observations. Given that HR links to NPP, soil C pools, and temperature, we recommend synergistically using datasets that reflect these variables (instead of identifying metrics in isolation). This could provide better model process evaluation if implemented in a larger benchmarking framework, such as the International Land Model Benchmarking project (ILAMB; Collier et al., 2018; Hoffman et al., 2017). Model development will be crucial in the next decade of carbon cycle research, but so will tools to test mechanistic understanding and elucidate a coherent picture of the land–atmosphere carbon response to a changing climate.

*Code and data availability.* NOAA Earth System Research Laboratory $CO_2$ measurements (Dlugokencky et al., 2016; ftp://aftp.cmdl.noaa.gov/data/trace_gases/co2/flask/surface/) and the Climatic Research Unit's gridded temperature product (University of East Anglia Climatic Research Unit, 2017; https://doi.org/10.5285/edf8febfdaad48abb2cbaf7d7e846a86) are publicly available online. CASA test-bed information and fluxes have been previously published in Wieder et al. (2018). GEOS-Chem $CO_2$ response data are available at the University of Michigan Library Deep Blue online repository (Basile et al., 2019; https://doi.org/10.7302/xjzc-xy05).

*Supplement.* The supplement related to this article is available online at: https://doi.org/10.5194/bg-17-1-2020-supplement.

*Author contributions.* SJB and GKA designed the research. WRW, MDH, and XL contributed model components. SJB conducted the analysis. All authors contributed to discussions. SJB, GKA, and WRW wrote the manuscript.

*Competing interests.* The authors declare that they have no conflict of interest.

*Acknowledgements.* We thank NOAA ESRL for providing observations of atmospheric $CO_2$. We thank the Climate Research Unit for their historically gridded temperature product.

*Financial support.* This research has been supported by the National Aeronautics and Space Administration (NASA ROSES Interdisciplinary Science (grant no. NNX17AK19G)) and the U.S. Department of Energy (RUBISCO Science Focus Area, DOE Regional and Global Model Analysis program). William R. Wieder and Melannie D. Hartman were also supported by grants from the U.S. Department of Agriculture, National Institute of Food and Agriculture award 2015-67003-23485, and the U.S. Department of Energy, Biological and Environmental Research awards TES DE-SC0014374 and BSS DE-SC0016364.

*Review statement.* This paper was edited by Martin De Kauwe and reviewed by three anonymous referees.

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

**Remarks from the language copy-editor**

CE1    Please confirm 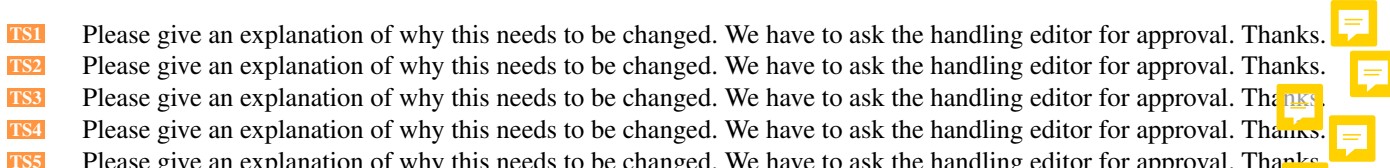

**Remarks from the typesetter**

TS1    Please give an explanation of why this needs to be changed. We have to ask the handling editor for approval. Thanks.

TS2    Please give an explanation of why this needs to be changed. We have to ask the handling editor for approval. Thanks.

TS3    Please give an explanation of why this needs to be changed. We have to ask the handling editor for approval. Thanks.

TS4    Please give an explanation of why this needs to be changed. We have to ask the handling editor for approval. Thanks.

TS5    Please give an explanation of why this needs to be changed. We have to ask the handling editor for approval. Thanks.