# Peer review of "Leveraging the signature of heterotrophic respiration on atmospheric CO2 for model benchmarking."

_Biogeosciences, 2019_

## Referee Comment (RC1) · Anonymous Referee #1 · 30 Jul 2019

This study uses globally modeled ecosystem CO2 fluxes and an atmospheric transport model together with observed atmospheric CO2 concentrations to evaluate different soil respiration models. This is done by comparing the inter-annual and intra-annual variability in CO2 fluxes between model and observations. The global scale of the study makes this a rather rough comparison, which however is useful for identifying soil models that perform better at such scales.

In general, there seems to be a considerable amount of work and careful analysis carried out and the study had scientifically interesting results. The manuscript has good grammar and few typos. The introduction goes from the more general to the

specific aims and summary of the methods and is well written. On the down side, much in the methods and results remains difficult to follow. Explaining the procedure in more detail will largely facilitate understanding the results and discussion. Generally, the text should be made more readable to a biogeochemistry audience that may not be familiar with the jargon and assumptions used in this specific field. Because there are many variables and procedures being used, it would help to more carefully define each part and consistently use the defined terms throughout the text. The manuscript should reach publication quality after these aspects are taken care of.

Specific remarks:

L44: is this not rather a "spatial" rather than "concentration" footprint?

L108: The term "phasing" is used in several places. Please describe its meaning on first use.

L114: remove "were"

L133-134: Please explain in more detail what you mean by diagnosing using one standard deviation.

L138-142: Six latitude zones are mentioned but 5 used later. Maybe clarify here that 2 zones are aggregated. Also, could you use the same naming for north and south (e.g. NHL, SHL, etc.).

L162-166: CLM can be run to simulate the mentioned fluxes. Why was CASA used instead?

L194: model structure?

L196-197: The procedure here is unclear. What do you mean by masked to land regions that align with the sampling zones? Do you mean the latitude zones defined earlier? Why do you specify "land regions", is it because there are ocean fluxes that need to be excluded? "mask" is a term that many reader will not understand. Why are

monthly averages calculated if daily values are used for the transport model? Please add detail.

L197-199: A repetition of the zones does not seem necessary.

L203-204: Please give more detail, either here or below, of how exactly these are used as boundary conditions for the model.

L204: Do you mean the transport model is run for lat zones separately? How does is deal with the lat boundaries in this case?

L211: "data" rather than "fields"

L215: Explain here the model CO2 flux inputs and outputs (presumably the boundary conditions mentioned before?) and clarify what fluxes are omitted (fossil fuels, etc)

L216: "minimize the"

L217: Confusing. Anthropogenic emissions are not modeled, so they cannot have an influence. Presumably you are relating this to the observations. Please reword.

L219: "closest to the observation sites, i.e. the"

L221: remove "calculations"

L222-225: This clarification should come before, when the averaging is explained. Although, as stated, atmospheric [CO2] over the ocean should integrate the signal over large regions, it is not clear why the zones and averaging was necessary. Since model data can be obtained for the observation points, the residuals of these could have been analyzed directly.

L226-242: This section is not clear. L226: "spatial" you mean the lat zones? How exactly do you isolate the imprints? What do you mean by "tag"? I believe "track CO2 tracers" is confusing since "tracer" relates to flow dynamics, but the analysis is about concentration changes(?). You mention "4 sets of fluxes" but aren't the observations

CO2 concentrations. What is then being compared (you mention CO2NEP variability is comparable to observations)? Are CO2NEP and CO2HR fluxes or concentrations? If these are the fluxes, should they not be introduced in the previous section?

L245-248: The calculation is not the variability but just the growth rate (or rate of change). Where does the annual IAV time series come from? Is it an annual average of the IAV series? Please clarify. Consider rewording, e.g. "The monthly and yearly rate of change of the the interannual anomaly (i.e. the IAV timeseries) was calculated as ...".

L248-250: This sentence is not understandable.

L251-253: "fitted to ... against ..." instead of "for .. with ...". Also, where does T IAV come from?

L255: "global temperature sensitivity"

L255-259: Was this reference calculated from IAV data or actual observations? What is this reference for?

L263-277: Global variability of what? What do you define as SDrel, the ratio or the "IAV magnitude"? (presumably the latter, but be more explicit). In any case, the term IAV magnitude is confusing. Why not simply say IAV relative SD? The calculation should be made clearer. What are "regional values of simulated CO2" (since all simulations made are regional CO2). Why do you use the relative standard deviation (or CV). A high CV from a small flux can have a smaller impact on global values than a small CV from a large flux. The actual SD may be a better measure. Also, it seems to me that using a ratio between regions and global assumes an additive effect of each region, which might not be the case (what if the regional IAV CV is larger than the global value?). What do you mean by sourced only from a single region? Consider rewriting this paragraph/revising this analysis.

L283-285: Still at a loss of what CO2NPP, CO2HR are. Atmospheric values derived

from...? Is it in any way logical that peaks in CO2NPP are max in April (early spring) and min August (mid summer) in NHL?

L299-300: Does CO2NEP not always consider NPP and HR together?

L338: Not if NPP increases more than HR.

Figure 2 and 3: If I understood correctly, values here are differences with the long term trend. To make this clearer, it would be good to note this in the caption.

---

## Referee Comment (RC2) · Anonymous Referee #2 · 30 Sep 2019

This paper connected heterotrophic respiration and atmospheric CO2 concentrations, and examined impact of signals of heterotrophic respiration on atmospheric CO2 using model outputs. This study suggests that the atmospheric CO2 concentrations could be used to verify the global estimation of heterotrophic respiration, and furthermore verify models. The idea is interesting and the manuscript is basically clear. I have just a couple of comments.

The diagram of your analysis (kind of flowchart) is necessary to understand this analysis.

Seasonal cycles of HR in Fig. 2: The peaks of HR in middle and high latitudes of

northern hemisphere are in autumn (Sep to Oct). Is this widely accepted? For me, the peaks should be in July-Sep. Are the heterotrophic respiration and NPP you used well correlated with observation data oriented estimates in terms of amount, seasonality, and spatial pattern? This is important for this study.

There seems several jargons which the authors should explain more carefully or replace them with easier wordings.

---

## Referee Comment (RC3) · Anonymous Referee #3 · 13 Nov 2019

Basile and colleagues compare three model formulations of heterotrophic respiration in their predictions of CO2 generation to the atmosphere, and compare the predictions with observations of atmospheric CO2 concentrations from a series of oceanic observations. I found the direct comparison of model formulations to be important and timely, given that the authors compared a CENTURY-like traditional formulation to more recent "mechanistic" models that explicitly simulate microbial processes.

In some ways, this is a well-written manuscript. The text has the crisp precision is a hallmark of good scientific writing. However, the manuscript is also challenging to understand and follow, in part because it uses jargon as well as many symbols and

acronyms. I suggest that the authors embed summary sentences at the end of some paragraphs throughout the results and discussion section to sum up the meaning of the results for the reader, without using acronyms (e.g., "These results suggest that the preponderance of the CO2 production driving the seasonal cycle of atmospheric CO2 originates in the southern tropical region").

Line 60. I agree with the central message of this paragraph, but I would further emphasize that HR is exceptionally challenging to measure, even at the local scale. Separating soil respiration into autotrophic and heterotrophic components is possible, and it has been done well in a few places where isotopic techniques are possible on intact soils, but it has also been done poorly or with significant limitations in other places. This is, in part, because of the intrinsic linkage between microbial decomposition and root activity (i.e., exudation, allocation of carbohydrate to mycorrhizal partners). I encourage the authors to acknowledge the uncertainty in estimating HR from soil respiration fluxes, similar to their statement regarding NEE measurements (∼line 64).

Line 80. I appreciate this text directly comparing models like CENTURY to the newer, "more mechanistic" models that explicitly simulate microbial processes. Directly comparing these modeling frameworks is timely and important.

Line 239. I am somewhat concerned by the lack of treatment of ocean CO2 fluxes, which are quantitatively large relative to the other fluxes listed here. I appreciate the following sentence, which at least partially addresses my concern. The authors might consider specifically state the assumption they are making by ignore these fluxes, which is that ocean CO2 fluxes are constant at seasonal and interannual timescales. This assumption is challenging to swallow, particularly given that the atmospheric CO2 observations were made in areas surrounded by oceans.

I found the ordering of the results to be challenging to understand. I first wanted to see an assessment of the model simulations relative to the data at the two temporal scales of interest here (seasonal and interannual). I did not find Figure 2 or it's associated

text at the beginning of the results section to be useful in aiding my understanding. I am likely missing something. However, I would find the results to be structured more understandably (for me) if the current figures 3 and 4 became the first figures presented as results. That is, the authors may consider omitting figure 2, or moving it down.

I was surprised by the relative lack of direct comparisons across these three models in the discussion section. I was hoping for more explicit "unpacking" of the particular model formulations, with direct recommendations as to which model components are most justifiable given the observed data. I found that much of the discussion amounted to throw-away sentences such as line 457-460, in which little of consequence was said regarding how we should model HR.

––––––––––––––––––––––––––––––––

---

## Author Comment (AC1) · 11 Jan 2020

This study uses globally modeled ecosystem CO2 fluxes and an atmospheric transport model together with observed atmospheric CO2 concentrations to evaluate different soil respiration models. This is done by comparing the inter-annual and intra-annual variability in CO2 fluxes between model and observations. The global scale of the study makes this a rather rough comparison, which however is useful for identifying soil models that perform better at such scales. In general, there seems to be a considerable amount of work and careful analysis carried out and the study had scientifically interesting results. The manuscript has good grammar and few typos. The introduction

goes from the more general to the specific aims and summary of the methods and is well written. On the down side, much in the methods and results remains difficult to follow. Explaining the procedure in more detail will largely facilitate understanding the results and discussion. Generally, the text should be made more readable to a biogeochemistry audience that may not be familiar with the jargon and assumptions used in this specific field. Because there are many variables and procedures being used, it would help to more carefully define each part and consistently use the defined terms throughout the text. The manuscript should reach publication quality after these aspects are taken care of.

We thank the reviewer for their time and comments. We appreciate there are elements of the paper that need more translation between the atmospheric and biogeochemical communities. In response to the reviewer's suggestions, we have clarified definitions and explanations throughout the paper. We will explicitly include the following text and equations in sections 2.1 and 2.3. We further address the reviewer's individual comments below.

Text and equations added to section 2.1:

"The testbed is a chain of model simulations where soil models with different structures can be run under the same forcing data, including the same gross primary productivity (GPP) fluxes, soil temperature, and soil moisture. The testbed produces its own estimates of net primary production (NPP), the difference between GPP and autotrophic respiration (AR; Eqn. 1). Each testbed soil model in this analysis produces unique gridded heterotrophic respiration (HR) values based on its own underlying mechanism and soil C stocks. ... From the testbed output we calculate the net ecosystem productivity (NEP; Eqn. 3). In the analysis presented here, CASA NPP was used across the testbed ensemble in the NEP calculation, thus highlighting differences in the timing and magnitude of HR fluxes from the individual soil models. From a land perspective (positive NEP fluxes into land), NEP is calculated as NPP – HR, where respiration release of $CO_2$ decreases net carbon gains through photosynthesis. Here, we use an atmospheric perspective for NEP (positive NEP fluxes into the atmosphere) by reversing the sign on the NPP flux and taking HR as positive (Eqn. 3)."

NPP=GPP-AR (1) NEP=HR-NPP (2)

... Text and equations added to section 2.3: "Throughout the manuscript, we refer to $CO_2$ originating from these NPP and HR component fluxes as CO2NPP and CO2HR, respectively. We use a sign convention for the fluxes whereby a positive value indicates a source of carbon to the atmosphere, which means we can combine the $CO_2$ tracers from NPP and HR to calculate the expected atmospheric variation owing to NEP using (Eqn. 3):

$CO\_2^{NEP}=CO\_2^{HR}+CO\_2^{NPP}$ (3)

Specific remarks: L44: is this not rather a "spatial" rather than "concentration" footprint?

We thank the reviewer for this comment, as concentration footprint is an atmospheric science term that may be less familiar to the biogeochemistry community. We have therefore changed the usage in the revised manuscript: "The high precision and accuracy of in situ observations and the fact that these measurements integrate information about ecosystem carbon fluxes over a large spatial footprint make atmospheric $CO_2$ a strong constraint on model predictions of net carbon exchange (Keppel-Aleks et al., 2013)."

L108: The term "phasing" is used in several places. Please describe its meaning on first use.

The term was updated in this line as the following: "For example, at seasonal timescales, improving the parameterization for litterfall in the CASA model improved phasing – i.e., the timing of seasonal maxima, minima, and inflection points – for the simulated atmospheric $CO_2$ annual cycle (Randerson et al., 1996)."

L114: remove "were"

The sentence was updated.

L133-134: Please explain in more detail what you mean by diagnosing using one standard deviation.

We have modified the text to describe the use of standard deviation to calculate the magnitude of interannual variability: "The magnitude of $CO_2$ IAV is calculated as one standard deviation on the detrended, deseasonalized timeseries, unless otherwise noted."

L138-142: Six latitude zones are mentioned but 5 used later. Maybe clarify here that 2 zones are aggregated. Also, could you use the same naming for north and south (e.g. NHL, SHL, etc.).

The distinction between latitude bands used for aggregating atmospheric $CO_2$ observations and the latitude bands used for the flux analysis is made in the revised manuscript at Lines 139 to 142. The hemisphere naming convention has been updated at Lines 140 to 142 to reflect the same abbreviations used later in the text: "Northern Hemisphere high latitudes (NHL: 61 to 90°N), midlatitudes (NML: 24 to 60°N), tropics (NT: 1 to 23°N), Southern Hemisphere tropics (ST: 0 to 23°S), and two Southern extratropics bands: the southern midlatitudes (SML, 24-60 S), and the southern high latitudes (SHL, 61-90 S). The global mean $CO_2$ timeseries is constructed as an area-weighted average of these six atmospheric zones."

L162-166: CLM can be run to simulate the mentioned fluxes. Why was CASA used instead?

The idea behind the testbed was to generate a more generalizable framework for model comparisons that could take observations (point scale simulation forced with flux tower data) or modeled output from land models like CLM, ELM, DAYCENT, etc and deliver consistent forcings to the soil models (CASA, MIMICS, CORPSE). This would allow us also to look at forcing uncertainty from other models (which is not the focus of the work

we presented). For these runs we chose to use GPP, soil moisture, soil temperature, etc from CLM simulations (CLM4.5-SP with Cru-NCEP forcing) as the input for the testbed, in part because this model configuration has been documented in Wieder et al., 2018. For the analysis presented in this paper, CASA NPP was used across the soil model ensemble in the NEP calculation for consistency in highlighting HR differences. The following sentence was added to the text for clarification in section 2.2: "In the analysis presented here, CASA NPP was used across the testbed ensemble in the NEP calculation, thus highlighting differences in the timing and magnitude of HR fluxes from the individual soil models."

L194: model structure?

We provide a definition of model structure in the updated text: "Overall, while the testbed approach contains necessary simplifications, it provides the ability to query the role of model structure, including assumptions about the number of soil carbon pools, the role of microorganisms, and the sensitivity to environmental factors, in driving HR flux differences when NPP and environmental controls are held in common."

L196-197: The procedure here is unclear. What do you mean by masked to land regions that align with the sampling zones? Do you mean the latitude zones defined earlier? Why do you specify "land regions", is it because there are ocean fluxes that need to be excluded? "mask" is a term that many reader will not understand. Why are monthly averages calculated if daily values are used for the transport model? Please add detail.

We have used the reviewer's comment as an opportunity to streamline the methods, and we have also added a flow diagram based on Reviewer 2's comment. The text has been updated as follows: "The testbed fluxes are used in two ways: first, we analyze monthly-averaged, regional fluxes for net primary production (NPP) from CASA-CNP and HR simulated by CASA-CNP, CORPSE and MIMICS. Second, we use the raw daily fluxes as boundary conditions for global GEOS-Chem runs to simulate the influence of

these fluxes on atmospheric CO2, as described in the following section."

L197-199: A repetition of the zones does not seem necessary.

This has been removed.

L203-204: Please give more detail, either here or below, of how exactly these are used as boundary conditions for the model.

Based on the reviewer's comment, we modified the order of this description to section 2.3 and provided the following details: "The testbed fluxes from 1980 to 2010 are used for land emissions to simulate the imprint of these different soil model configurations on atmospheric CO2 (Fig. 1). In our simulations, HR and NPP fluxes were separated into the five regions listed above (NHL, NML, NT, ST, SE) so that the influence of carbon fluxes originating from these individual regions on global atmospheric CO2 mole fraction could be quantified. We initialized separate species of CO2 in the atmospheric model, one for each flux (HR or NPP) and region (NHL, NML, etc.). Since we considered four fluxes (CASA NPP and three types of HR) originating in five regions, we simulated a total of 20 species. These species were tracked throughout the simulation as their spatiotemporal distribution changed due to the combined influence of CO2 fluxes at the surface and atmospheric weather. Although these species are simulated individually, we can simply sum the regional atmospheric species for a given flux (e.g., CASA HR) to determine the atmospheric CO2 arising from all fluxes over the globe."

L204: Do you mean the transport model is run for lat zones separately? How does is deal with the lat boundaries in this case?

Simulations are all run globally, so there is no issue with latitudinal boundary conditions. The text has been updated in the previous comment to explain the use of land masking to isolated latitudes from each other in the emissions component of the model simulation.

L211: "data" rather than "fields"

This change was made in the revised manuscript.

L215: Explain here the model CO2 flux inputs and outputs (presumably the boundary conditions mentioned before?) and clarify what fluxes are omitted (fossil fuels, etc)

We updated the text to reflect that we did simulate the contribution of ocean and fossil CO2, but neglect the CO2 mole fraction owing to these fluxes: "We also simulated the fossil and ocean imprint on atmospheric CO2 using boundary conditions from CO2 CAMS inversion 17r1 (https://atmosphere.copernicus.eu/sites/default/files/2018-10/CAMS73_2015SC3_D73.1.4.2-1979-2017-v1_201807_v1-1.pdf). However, at the temporal scales of this analysis, ocean and fossil fuel fluxes had a much smaller influence on regional patterns of atmospheric CO2 than did land fluxes. Across the six latitude bands, the detrended CO2NEP annual amplitude ranges from a factor of 1.5 (in the tropics) to an order of magnitude larger (at high latitudes) than CO2 from ocean fluxes and fossil fuel emissions. Likewise, the IAV from fossil and ocean-derived CO2 was at most 25% that of NEP-derived CO2 at most latitude bands. These results are consistent with previous studies that have demonstrated that NEP drives most of the atmospheric CO2 seasonality (> 90%; Nevison et al., 2008; Randerson et al., 1997) and interannual variability (e.g., Rayner et al. 2008; Battel et al. 2000). Given that patterns of IAV in ocean and fossil CO2 partially cancel each other and the large uncertainty in ocean fluxes, we choose to omit these CO2 species from our analysis."

L216: "minimize the"

The text has been updated following this and the next comment.

L217: Confusing. Anthropogenic emissions are not modeled, so they cannot have an influence. Presumably you are relating this to the observations. Please reword.

The reviewer is correct that this was done to improve the utility of atmospheric observations as a benchmark for the simulations, and we have revised the text to read: "We sample the gridded atmospheric simulation output at the 34 marine boundary layer

(MBL) sites identified in section 2.1, using the 3rd vertical level to minimize influence of land-atmosphere boundary layer dynamics. We then calculate the latitude zone average, median annual cycle and interannual variability using the methods described for CO2 observations (see section 2.1). Averaging CO2 from all sites within a latitude band is consistent with our hypothesis that atmospheric CO2 may provide constraints on large-scale patterns of heterotrophic respiration, but individual sites may be too heavily influenced by local characteristics not accounted for by the testbed fluxes. As such, averaging simulated and observed CO2 across latitude zones smooths local information while retaining information about regional scale fluxes."

L219: "closest to the observation sites, i.e. the"

Wording correction was addressed in response to the previous comment.

L221: remove "calculations"

Text was removed.

L222-225: This clarification should come before, when the averaging is explained. Although, as stated, atmospheric [CO2] over the ocean should integrate the signal over large regions, it is not clear why the zones and averaging was necessary. Since model data can be obtained for the observation points, the residuals of these could have been analyzed directly.

Given the generalized framework of the testbed, and the fact that it doesn't account for highly local processes that might affect the CO2 observations, we prefer to use the zonal averaging approach we outlined rather than compare with individual sites. However, we have added clarification as the reviewer suggested: "Averaging CO2 from all sites within a latitude band is consistent with our hypothesis that atmospheric CO2 may provide constraints on large-scale patterns of heterotrophic respiration, but individual sites may be too heavily influenced by local characteristics not accounted for by the testbed fluxes. As such, averaging simulated and observed CO2 across latitude zones

smooths local information while retaining information about regional scale fluxes."

L226-242: This section is not clear.

Text changes in section 2.2 have expanded the explanation of NPP and HR as boundary conditions for the GEOS Chem atmospheric transport model simulations. A figure demonstrating the land masking technique used has been added to the Supplementary information as well as a chart detailing the input fluxes and resulting CO2 naming convention. Additionally, Line 242 has been moved to section 2.2 to explain the use of NEP as a variable with the accompanying equation defining the term.

L226: "spatial" you mean the lat zones? How exactly do you isolate the imprints? What do you mean by "tag"? I believe "track CO2 tracers" is confusing since "tracer" relates to flow dynamics, but the analysis is about concentration changes(?). You mention "4 sets of fluxes" but aren't the observations CO2 concentrations. What is then being compared (you mention CO2NEP variability is comparable to observations)? Are CO2NEP and CO2HR fluxes or concentrations? If these are the fluxes, should they not be introduced in the previous section?

We thank the reviewer for this reminder to remove atmospheric science jargon from the paper. We have replaced the use of the word "tracer" with "species" throughout to indicate that we consider each region to affect only one CO2 species. The concentration of each species is then directly tied back to a single flux. The updated text reads: "The testbed fluxes from 1980 to 2010 are used for land emissions to simulate the imprint of these different soil model configurations on atmospheric CO2 (Fig. 1). In our simulations, HR and NPP fluxes were separated into the five regions listed above (NHL, NML, NT, ST, SE) so that the influence of carbon fluxes originating from these individual regions on global atmospheric CO2 mole fraction could be quantified. We initialized separate species of CO2 in the atmospheric model, one for each flux (HR or NPP) and region (NHL, NML, etc.). Since we considered four fluxes (CASA NPP and three types of HR) originating in five regions, we simulated a total of 20 species. These species

were tracked throughout the simulation as their spatiotemporal distribution changed due to the combined influence of $CO_2$ fluxes at the surface and atmospheric weather. Although these species are simulated individually, we can simply sum the regional atmospheric species for a given flux (e.g., CASA HR) to determine the atmospheric $CO_2$ arising from all fluxes over the globe." All modeled $CO_2$ is output as concentrations (ppm) and the ESRL $CO_2$ observations are also reported as mole fraction (ppm). Additional wording was added to section 2 to clarify variable units: Lines 133 to 135: "For this analysis we use reference $CO_2$ measurements reported in parts per million (ppm) from 34 marine boundary layer sites (MBL, Table S1) within the NOAA Earth System Research Laboratory sampling network (ESRL, Fig. 1; Dlugokencky et al., 2016)." Lines 156 to 157: "All testbed fluxes are output in grams of carbon per meter square at a daily temporal resolution and then converted to petagrams (Pg C, over a region)." Lines 227 to 228: "The model is initialized with globally-uniform atmospheric $CO_2$ mole fraction equal to 350 ppm."

L245-248: The calculation is not the variability but just the growth rate (or rate of change). Where does the annual IAV time series come from? Is it an annual average of the IAV series? Please clarify. Consider rewording, e.g. "The monthly and yearly rate of change of the the interannual anomaly (i.e. the IAV timeseries) was calculated as ...".

The text was reworded to explain the calculation of the growth rate anomaly (rate of change in the interannual $CO_2$ values, or residual values after detrending and removing the seasonal cycle) which is a distinct value from the growth rate (rate of change in the unaltered $CO_2$ values): "Rates of change were derived from monthly and annual timeseries to ultimately calculate the temperature sensitivity of the testbed fluxes, the modeled $CO_2$, and the observed $CO_2$ values. The $CO_2$ growth rate anomaly was calculated as the difference between timestep n and n-1 in both the monthly and annual $CO_2$ IAV timeseries. As a result of this technique, the monthly $CO_2$ growth rate anomalies were centered on the first day of the corresponding months. To compare

flux information with CO2 growth rate anomalies, daily testbed flux timeseries were averaged to monthly resolution and then interpolated by averaging between months to center values on the first day of each month."

L248-250: This sentence is not understandable.

This wording was replaced in the response to the previous comment.

L251-253: "fitted to ... against ..." instead of "for .. with ...". Also, where does T IAV come from?

The text has been reworded as: "Following Arora et al. (2013), we calculate temperature sensitivity ($\gamma$) using an ordinary linear regression (OLR). We calculate OLR for the interannual variability timeseries of CASA-CNP soil temperature (T IAV) against 1) atmospheric CO2 growth rate anomalies, and 2) land flux IAV (see section 2.2)."

L255: "global temperature sensitivity"

The text has been updated.

L255-259: Was this reference calculated from IAV data or actual observations? What is this reference for?

The term "reference" was used needlessly in the original manuscript, and the text has been updated as: "The global temperature sensitivity value for the observed CO2 growth rate anomaly was calculated for 1982 to 2010 using ESRL CO2 observations and the Climatic Research Unit's gridded temperature product (CRU TS4; Jones et al., 2012), which is derived from interpolated ground station measurements."

L263-277: Global variability of what? What do you define as SDrel, the ratio or the "IAV magnitude"? (presumably the latter, but be more explicit). In any case, the term IAV magnitude is confusing. Why not simply say IAV relative SD? The calculation should be made clearer. What are "regional values of simulated CO2" (since all simulations made are regional CO2). Why do you use the relative standard deviation (or CV). A

high CV from a small flux can have a smaller impact on global values than a small CV from a large flux. The actual SD may be a better measure. Also, it seems to me that using a ratio between regions and global assumes an additive effect of each region, which might not be the case (what if the regional IAV CV is larger than the global value?). What do you mean by sourced only from a single region? Consider rewriting this paragraph/revising this analysis.

We thank the reviewer for this comment and the opportunity to clarify the methods we used. For the flux analysis, the SDrel is the ratio of the IAV magnitude in a given region relative to the IAV magnitude of the globe (five regions summed). For the CO2 analysis, the SDrel is the ratio of the IAV magnitude of the global-average atmospheric CO2 arising from a single region relative to the IAV magnitude from the global fluxes. Thus, we are not using a coefficient of variability, which would normalize the magnitude of IAV to the baseline flux. As the reviewer notes, we avoid this formulation because it could lead to a high CV when fluxes are small, which is undesirable. We agree with the reviewer that the SDrel from the five regions is not necessarily additive, which is why we have provided correlation coefficients between the regional and global information since not all variations will be coherent with the global signal. The text has been made clearer as to why the values were chosen and how they are defined "We also assess the influence of individual regions on the global mean signal for both component land fluxes (NPP, HR) and simulated atmospheric CO2 (CO2NPP, CO2HR, CO2NEP). We first quantify the magnitude of variability in each region relative to the magnitude of global variability ($\sigma$REL) as the ratio of regional IAV standard deviation to global IAV standard deviation. This ratio is calculated for monthly flux IAV from each of the five flux regions and for the global-mean CO2 timeseries that arises from fluxes in each of the five flux regions (e.g., the global CO2 response to NHL fluxes, or the global CO2 response to NML fluxes, etc.). The value of $\sigma$REL has a lower bound of 0, which would indicate that a region contributes no IAV, but has no upper bound, since a value greater than 1 simply indicates that the fluxes in a given region are more variable than global fluxes. We note that the timing of IAV in a given region may be independent of IAV

in other regions, and thus may or may not be temporally in-phase with global IAV. We therefore also calculate correlation coefficients (r) for the timeseries of regional flux IAV and CO2 IAV with the global signal. Thus, if an individual region were responsible for all observed global flux or CO2 variability, it would have both $\sigma$REL and r values equal to 1 in this comparison. The value for r will be small if a regional signal is not temporally coherent with the global signal, even if the magnitude of variability is high."

L283-285: Still at a loss of what CO2NPP, CO2HR are. Atmospheric values derived from...? Is it in any way logical that peaks in CO2NPP are max in April (early spring) and min August (mid summer) in NHL?

Definitions for CO2NPP, CO2HR are included in section 2.3. Also the peaks relate to the reversal in the sign of the NPP flux, which carries through to CO2NPP. Descriptive wording explaining the sign convention have been added to the text and Figure 2 (now Figure 4) caption. The text has been updated as: "Within the modeled carbon dioxide concentrations resulting from land fluxes, CO2NPP and CO2HR, show largest season-ality in the NHL, with seasonal amplitudes decaying toward the tropics and Southern Hemisphere. In the NHL, the peak-to-trough amplitude of CO2NPP is 39±2 ppm, with a seasonal maximum in April and a seasonal minimum in August (Fig. 4a; note, this CO2NPP peak reflects the sign reversal in the driving NPP flux (section 2.3))."

L299-300: Does CO2NEP not always consider NPP and HR together?

The reviewer is correct that CO2NEP always considers NPP and HR together, and equations have been added to section 2 to make this relationship clear.

L338: Not if NPP increases more than HR.

The text has been updated for clarification as: "Both global NPP and HR fluxes are sensitive to temperature variations at interannual timescales, with increased build-up of CO2 in the atmosphere at higher temperatures, in part because the rate of HR increases at higher temperature and in part because most latitude bands show a reduction in NPP at above-average temperatures."

Figure 2 and 3: If I understood correctly, values here are differences with the long term trend. To make this clearer, it would be good to note this in the caption.

Clarification that the seasonal cycles shown are detrended values has been added to the figure captions.

Please also note the supplement to this comment:
https://www.biogeosciences-discuss.net/bg-2019-256/bg-2019-256-AC1-supplement.pdf

[Figure]

**Fig. 1.** Figure 1: Flow chart depiction of the analysis process from soil model fluxes to simulated CO2 concentration and comparison with NOAA observations.

[Figure]

**Fig. 2.** Figure 2: Tagged flux regions and marine boundary layer CO2 observing sites used in our analysis. The 5 tagged flux regions are shown in color fill: Northern High Latitude (NHL), Northern Mid-Latitude

[Figure]

**Fig. 3.** Figure 3: Climatological annual cycle (median) of CO2 for observations (black) and global net ecosystem productivity flux (CO2NEP, colors) between 1982 and 2010. Monthly climatology values were create

[Figure]

**Fig. 4.** Figure 4: Climatological annual cycle (median) of atmospheric CO2 simulated from land fluxes (CO2NPP, CO2HR) between 1982 and 2010. Monthly climatology values were created after detrending the CO2 tim

[Figure]

**Fig. 5.** Figure 5: Interannual variability of CO2 from global net ecosystem productivity (CO2NEP IAV) for testbed models (colors) and marine boundary layer observations from the NOAA ESRL network (black). Gra

[Figure]

[Figure]

**Fig. 6.** Figure 6: Magnitude of CO2 interannual variability resulting from (a) individual flux components (CO2NPP IAV, CO2HR IAV) and (b) global net ecosystem productivity (CO2NEP IAV). Observed CO2 IAV from N

[Figure]

**Fig. 7.** Figure 7: Temperature sensitivity ($\gamma$) calculated for interannual variability (IAV) of CASA-CNP air temperature and (a) flux IAV and corresponding CO2 growth rate anomalies, (b) NEP IAV and CO2NEP grow

[Figure]

**Fig. 8.** Figure 8: Comparison of regional and global interannual variability (IAV) from land fluxes and resulting atmospheric CO2 between 1982 and 2010. (a, c, e) Normalized ratio taken between regional IAV an

**Supplement:**

Table S1: Marine Boundary Layer (MBL) stations within the NOAA Earth System Research Laboratory $CO_2$ sampling network (ESRL). These sites were selected for obtaining at least 50% data coverage over the analysis period of 1982 to 2010.

| Region | Station | Acronym | Lat | Lon |
|---|---|---|---|---|
| 60°–90°N | Alert, AK | ALT | 82.5 | -62.5 |
| | Ny-Ålesund, Svalbard | ZEP | 78.9 | 11.9 |
| | Barrow, AK | BRW | 71.3 | -156.6 |
| | Stórhöfði, Iceland | ICE | 63.4 | -20.0 |
| 23°–60°N | Mace Head, Ireland | MHD | 53.3 | -9.9 |
| | Shemya, AK | SHM | 52.7 | 174.1 |
| | Terceira Island, Azores | AZR | 38.8 | -27.4 |
| | Tudor Hill, Bermuda | BMW | 32.3 | -64.9 |
| | Sand Island, Midway | MID | 28.2 | -177.4 |
| | Key Biscayne, FL | KEY | 25.7 | -80.2 |
| | Pacific Ocean, 25°N | POCN25 | 25.0 | -135.0 |
| 0°–23°N | Pacific Ocean, 20°N | POCN20 | 20.0 | -139.0 |
| | Cape Kumukahi, HI | KUM | 19.5 | -154.8 |
| | Pacific Ocean, 15°N | POCN15 | 15.0 | -143.0 |
| | Mariana Islands, Guam | GMI | 13.4 | 144.8 |
| | Ragged Point, Barbados | RPB | 13.2 | -59.4 |
| | Pacific Ocean, 10°N | POCN10 | 10.0 | -147.0 |
| | Pacific Ocean, 5°N | POCN05 | 5.0 | -151.0 |
| | Christmas Island | CHR | 1.7 | -157.2 |
| 0°–23°S | Seychelles | SEY | -4.7 | 55.2 |
| | Pacific Ocean 5°S | POCS05 | -5.0 | -159.0 |
| | Ascension Island | ASC | -8.0 | -14.4 |
| | Pacific Ocean 10°S | POCS10 | -10.0 | -163.0 |
| | Tutuila American Samoa | SMO | -14.2 | -170.6 |
| | Pacific Ocean 15°S | POCS15 | -15.0 | -167.0 |
| | Pacific Ocean 20°S | POCS20 | -20.0 | -171.0 |
| 23°–60°S | Pacific Ocean 25°S | POCS25 | -25.0 | -174.0 |
| | Pacific Ocean 30°S | POCS30 | -30.0 | -177.0 |
| | Cape Grim, Australia | CGO | -40.7 | 144.7 |
| | Crozet Island | CRZ | -46.5 | 51.9 |
| 60°–90°S | Palmer Station, Antarctica | PSA | -64.0 | -64.0 |
| | Syowa Antarctica | SYO | -69.0 | 39.6 |
| | Halley Bay, Antarctica | HBA | -75.6 | -26.5 |
| | South Pole | SPO | -90.0 | -24.8 |

Table S2: Coefficient of variation for flux variables by latitude zone. All variables have been detrended using a third-order polynomial fit. For NEP, a negative sign represents flux into land and a positive sign represents a flux to the atmosphere from land.

| Region | Model Flux | Mean Flux

[Pg C yr$^{-1}$] | Flux Standard Deviation (STD)

[Pg C yr$^{-1}$] | STD : Flux

[%] |
|---|---|---|---|---|
| | CASA HR | 3.94 | 0.08 | 2 |
| 61°-90°N | CORPSE HR | 4.52 | 0.23 | 5 |
| | MIMICS HR | 3.96 | 0.09 | 2 |
| | CASA NPP | 4.07 | 0.09 | 2 |
| | CASA NEP | -0.13 | 0.05 | 40 |
| | CORPSE NEP | 0.45 | 0.16 | 35 |
| | MIMICS NEP | -0.11 | 0.08 | 76 |
| | | | | 0 |
| | CASA HR | 22.73 | 0.24 | 1 |
| 24°-60°N | CORPSE HR | 23.06 | 0.31 | 1 |
| | MIMICS HR | 22.88 | 0.48 | 2 |
| | CASA NPP | 23.15 | 0.28 | 1 |
| | CASA NEP | -0.42 | 0.40 | 96 |
| | CORPSE NEP | -0.09 | 0.46 | 505 |
| | MIMICS NEP | -0.27 | 0.66 | 245 |
| | | | | 0 |
| | CASA HR | 10.63 | 0.08 | 1 |
| 1°-23°N | CORPSE HR | 10.63 | 0.10 | 1 |
| | MIMICS HR | 10.57 | 0.22 | 2 |
| | CASA NPP | 10.66 | 0.44 | 4 |
| | CASA NEP | -0.03 | 0.48 | 1428 |
| | CORPSE NEP | -0.03 | 0.47 | 1571 |
| | MIMICS NEP | -0.09 | 0.62 | 732 |
| | | | | 0 |
| | CASA HR | 14.26 | 0.10 | 1 |
| 0°-23°S | CORPSE HR | 14.30 | 0.12 | 1 |
| | MIMICS HR | 14.24 | 0.29 | 2 |
| | CASA NPP | 14.52 | 0.57 | 4 |
| | CASA NEP | -0.27 | 0.61 | 232 |
| | CORPSE NEP | -0.22 | 0.58 | 262 |
| | MIMICS NEP | -0.28 | 0.84 | 296 |
| | | | | 0 |
| | CASA HR | 3.72 | 0.05 | 1 |
| 24°-90°N | CORPSE HR | 3.74 | 0.07 | 2 |
| | MIMICS HR | 3.74 | 0.08 | 2 |
| | CASA NPP | 3.77 | 0.26 | 7 |

| | | | |
|---|---|---|---|
| CASA NEP | -0.05 | 0.26 | 518 |
| CORPSE NEP | -0.03 | 0.24 | 810 |
| MIMICS NEP | -0.03 | 0.33 | 1135 |

Table S3: Multiple linear regression coefficients ($\gamma$) and $R^2$ are used to model interannual variability in heterotrophic respiration as a function of interannual variability in temperature,

NPP, or preceding year NPP. All variables have been detrended and deseasonalized. We list statistically significant predictors of HR IAV, as determined by p-values from ANOVA.

| Region | Model IAV | HR IAV Regression $\gamma^*$ , $R^2$ | | |
|---|---|---|---|---|
| | | CASA-CNP Temperature IAV [Pg C y$^{-1}$ K$^{-1}$] | CASA-CNP NPP Current year IAV [--] | CASA-CNP NPP Preceding year IAV [--] |
| 61°-90°N | CASA HR | **0.16** , 0.64 | **0.74** , 0.67 | -0.14, 0.02 |
| | CORPSE HR | **0.42**, 0.54 | **2.22**, 0.77 | -0.23, 0.01 |
| | MIMICS HR | **0.13**, 0.32 | **0.63**, 0.40 | 0.06, 0.00 |
| | CASA-CNP NPP | **0.15**, 0.43 | | |
| 24°-60°N | CASA HR | **0.78**, 0.58 | 0.2, 0.05 | 0.33, 0.15 |
| | CORPSE HR | **1.00**, 0.57 | -0.28, 0.06 | 0.32, 0.08 |
| | MIMICS HR | **1.74**, 0.70 | **-0.90**, 0.27 | 0.05, 0.00 |
| | CASA-CNP NPP | -0.38, 0.10 | | |
| 1°-23°N | CASA HR | 0.17, 0.14 | **-0.10**, 0.26 | **0.12**, 0.45 |
| | CORPSE HR | 0.00, 0.00 | -0.06, 0.07 | **0.17**, 0.61 |
| | MIMICS HR | **1.03**, 0.68 | **-0.40**, 0.60 | -0.03, 0.00 |
| | CASA-CNP NPP | **-1.57**, 0.42 | | |
| 0°-23°S | CASA HR | **0.24**, 0.17 | **-0.07**, 0.17 | **0.12**, 0.41 |
| | CORPSE HR | -0.01, 0.00 | -0.01, 0.00 | **0.17**, 0.61 |
| | MIMICS HR | **1.50**, 0.87 | **-0.46**, 0.79 | 0.03, 0.00 |
| | CASA-CNP NPP | **-2.65**, 0.72 | | |
| 24°-90°N | CASA | 0.00, 0.00 | 0.03, 0.03 | **0.12**, 0.32 |
| | CORPSE | -0.07, 0.04 | 0.09, 0.13 | **0.17**, 0.36 |
| | MIMICS | **0.32**, 0.42 | **-0.26**, 0.65 | -0.08, 0.04 |
| | CASA-CNP NPP | **-1.11**, 0.52 | | |

*bolded values are statistically significant (p < 0.05)

Figure S1: Depiction of interannual variability (IAV) calculation. (a) Multi-site mean CASA-CNP $CO_2^{NEP}$ in the Northern Hemisphere high latitudes (NHL) region for 1982 to 2010 ($CO_2^{NEP}$ = $CO_2^{HR}$ + $CO_2^{NPP}$). (b) Detrended CASA-CNP $CO_2^{NEP}$ timeseries after removing a third-order polynomit fit. (c) Climatological annual cycle calculated using the median of monthly values over the analysis period. (d) CASA-CNP $CO_2^{NEP}$ interannual variability calculated from removing the climatological annual cycle from each year in the detrended timeseries.

Figure S2: $CO_2$ concentrations from ocean fluxes. $CO_2$ output from GEOS Chem simulation forced by $CO_2$ CAMS inversion 17r1. (a) Averaged $CO_2$ in the Northern Hemisphere high latitudes (NHL) region for 1982 to 2010. (b) Detrended $CO_2$ timeseries after removing a third-order polynomit fit. (c) Climatological annual cycle calculated using the median of monthly values over the analysis period. (d) $CO_2$ interannual variability calculated from removing the climatological annual cycle from each year in the detrended timeseries.

Figure S3: $CO_2$ concentrations from fossil fuel emissions. $CO_2$ output from GEOS Chem simulation forced by $CO_2$ CAMS inversion 17r1. (a) Averaged $CO_2$ in the Northern Hemisphere high latitudes (NHL) region for 1982 to 2010. (b) Detrended $CO_2$ timeseries after removing a third-order polynomit fit. (c) Climatological annual cycle calculated using the median of monthly values over the analysis period. (d) $CO_2$ interannual variability calculated from removing the climatological annual cycle from each year in the detrended timeseries.

**Land mask Methodology: CASA Soil Heterotrophic Respiration (HR)**

Figure S4: Example land mask for the Northern Hemisphere High Latitudes (NHL). (a) CASA-CNP soil heterotrophic respiration (HR) for January 1982. (b) Global land mask with land gridcells set to a constant fill value and ocean gridcells set to NaN. (c) Land mask after application of a separate NHL latitude band mask with values of 0 south of 61 °N. Land gridcells were reassigned a value of 1. (d) NHL land mask weighted by area with Greenland removed.

---

## Author Comment (AC2) · 11 Jan 2020

This paper connected heterotrophic respiration and atmospheric CO2 concentrations, and examined impact of signals of heterotrophic respiration on atmospheric CO2 using model outputs. This study suggests that the atmospheric CO2 concentrations could be used to verify the global estimation of heterotrophic respiration, and furthermore verify models. The idea is interesting and the manuscript is basically clear. I have just a couple of comments.

We thank the reviewer for their time and comments.

[Figure]

The diagram of your analysis (kind of flowchart) is necessary to understand this analysis.

We appreciate the need for clarification in this interdisciplinary work and have added a new Figure 1 flow chart to the figure list, along with an introductory description in section 2. Figure 1 shows a flow chart depiction of the analysis process from soil model fluxes to simulated $CO_2$ concentration and comparison with NOAA observations.

Seasonal cycles of HR in Fig. 2: The peaks of HR in middle and high latitudes of northern hemisphere are in autumn (Sep to Oct). Is this widely accepted? For me, the peaks should be in July-Sep. Are the heterotrophic respiration and NPP you used well correlated with observation data oriented estimates in terms of amount, seasonality, and spatial pattern? This is important for this study.

The HR fluxes have been evaluated in Wieder et al., 2018. Figure 2 (now Figure 4) shows the seasonal cycle of atmospheric $CO_2$ arising from HR fluxes, thus showing the cumulative imprint of HR fluxes. While the HR flux might peak in July-Sep, the $CO_2$ concentration from HR continues to accumulate in the atmosphere leading to the peak in autumn.

There seems several jargons which the authors should explain more carefully or replace them with easier wordings.

We thank the reviewer for this note and have made a conscientious effort throughout the revisions to add additional explanation and remove jargon.

Please also note the supplement to this comment:
https://www.biogeosciences-discuss.net/bg-2019-256/bg-2019-256-AC2-supplement.pdf

―――――――――――――――――――――

[Figure]

**Fig. 1.** Figure 1: Flow chart depiction of the analysis process from soil model fluxes to simulated CO2 concentration and comparison with NOAA observations.

[Figure]

**Fig. 2.** Figure 2: Tagged flux regions and marine boundary layer CO2 observing sites used in our analysis. The 5 tagged flux regions are shown in color fill: Northern High Latitude (NHL), Northern Mid-Latitude

[Figure]

**Fig. 3.** Figure 3: Climatological annual cycle (median) of CO2 for observations (black) and global net ecosystem productivity flux (CO2NEP, colors) between 1982 and 2010. Monthly climatology values were create

[Figure]

**Fig. 4.** Figure 4: Climatological annual cycle (median) of atmospheric CO2 simulated from land fluxes (CO2NPP, CO2HR) between 1982 and 2010. Monthly climatology values were created after detrending the CO2 tim

[Figure]

**Fig. 5.** Figure 5: Interannual variability of CO2 from global net ecosystem productivity (CO2NEP IAV) for testbed models (colors) and marine boundary layer observations from the NOAA ESRL network (black). Gra

[Figure]

[Figure]

**Fig. 6.** Figure 6: Magnitude of CO2 interannual variability resulting from (a) individual flux components (CO2NPP IAV, CO2HR IAV) and (b) global net ecosystem productivity (CO2NEP IAV). Observed CO2 IAV from N

[Figure]

**Fig. 7.** Figure 7: Temperature sensitivity ($\gamma$) calculated for interannual variability (IAV) of CASA-CNP air temperature and (a) flux IAV and corresponding CO2 growth rate anomalies, (b) NEP IAV and CO2NEP grow

[Figure]

**Fig. 8.** Figure 8: Comparison of regional and global interannual variability (IAV) from land fluxes and resulting atmospheric CO2 between 1982 and 2010. (a, c, e) Normalized ratio taken between regional IAV an

**Supplement:**

Table S1: Marine Boundary Layer (MBL) stations within the NOAA Earth System Research Laboratory $CO_2$ sampling network (ESRL). These sites were selected for obtaining at least 50% data coverage over the analysis period of 1982 to 2010.

| Region | Station | Acronym | Lat | Lon |
|---|---|---|---|---|
| 60°–90°N | Alert, AK | ALT | 82.5 | -62.5 |
| | Ny-Ålesund, Svalbard | ZEP | 78.9 | 11.9 |
| | Barrow, AK | BRW | 71.3 | -156.6 |
| | Stórhöfði, Iceland | ICE | 63.4 | -20.0 |
| 23°–60°N | Mace Head, Ireland | MHD | 53.3 | -9.9 |
| | Shemya, AK | SHM | 52.7 | 174.1 |
| | Terceira Island, Azores | AZR | 38.8 | -27.4 |
| | Tudor Hill, Bermuda | BMW | 32.3 | -64.9 |
| | Sand Island, Midway | MID | 28.2 | -177.4 |
| | Key Biscayne, FL | KEY | 25.7 | -80.2 |
| | Pacific Ocean, 25°N | POCN25 | 25.0 | -135.0 |
| 0°–23°N | Pacific Ocean, 20°N | POCN20 | 20.0 | -139.0 |
| | Cape Kumukahi, HI | KUM | 19.5 | -154.8 |
| | Pacific Ocean, 15°N | POCN15 | 15.0 | -143.0 |
| | Mariana Islands, Guam | GMI | 13.4 | 144.8 |
| | Ragged Point, Barbados | RPB | 13.2 | -59.4 |
| | Pacific Ocean, 10°N | POCN10 | 10.0 | -147.0 |
| | Pacific Ocean, 5°N | POCN05 | 5.0 | -151.0 |
| | Christmas Island | CHR | 1.7 | -157.2 |
| 0°–23°S | Seychelles | SEY | -4.7 | 55.2 |
| | Pacific Ocean 5°S | POCS05 | -5.0 | -159.0 |
| | Ascension Island | ASC | -8.0 | -14.4 |
| | Pacific Ocean 10°S | POCS10 | -10.0 | -163.0 |
| | Tutuila American Samoa | SMO | -14.2 | -170.6 |
| | Pacific Ocean 15°S | POCS15 | -15.0 | -167.0 |
| | Pacific Ocean 20°S | POCS20 | -20.0 | -171.0 |
| 23°–60°S | Pacific Ocean 25°S | POCS25 | -25.0 | -174.0 |
| | Pacific Ocean 30°S | POCS30 | -30.0 | -177.0 |
| | Cape Grim, Australia | CGO | -40.7 | 144.7 |
| | Crozet Island | CRZ | -46.5 | 51.9 |
| 60°–90°S | Palmer Station, Antarctica | PSA | -64.0 | -64.0 |
| | Syowa Antarctica | SYO | -69.0 | 39.6 |
| | Halley Bay, Antarctica | HBA | -75.6 | -26.5 |
| | South Pole | SPO | -90.0 | -24.8 |

Table S2: Coefficient of variation for flux variables by latitude zone. All variables have been detrended using a third-order polynomial fit. For NEP, a negative sign represents flux into land and a positive sign represents a flux to the atmosphere from land.

| Region | Model Flux | Mean Flux

[Pg C yr$^{-1}$] | Flux Standard Deviation (STD)

[Pg C yr$^{-1}$] | STD : Flux

[%] |
|---|---|---|---|---|
| | CASA HR | 3.94 | 0.08 | 2 |
| 61°-90°N | CORPSE HR | 4.52 | 0.23 | 5 |
| | MIMICS HR | 3.96 | 0.09 | 2 |
| | CASA NPP | 4.07 | 0.09 | 2 |
| | CASA NEP | -0.13 | 0.05 | 40 |
| | CORPSE NEP | 0.45 | 0.16 | 35 |
| | MIMICS NEP | -0.11 | 0.08 | 76 |
| | | | | 0 |
| | CASA HR | 22.73 | 0.24 | 1 |
| 24°-60°N | CORPSE HR | 23.06 | 0.31 | 1 |
| | MIMICS HR | 22.88 | 0.48 | 2 |
| | CASA NPP | 23.15 | 0.28 | 1 |
| | CASA NEP | -0.42 | 0.40 | 96 |
| | CORPSE NEP | -0.09 | 0.46 | 505 |
| | MIMICS NEP | -0.27 | 0.66 | 245 |
| | | | | 0 |
| | CASA HR | 10.63 | 0.08 | 1 |
| 1°-23°N | CORPSE HR | 10.63 | 0.10 | 1 |
| | MIMICS HR | 10.57 | 0.22 | 2 |
| | CASA NPP | 10.66 | 0.44 | 4 |
| | CASA NEP | -0.03 | 0.48 | 1428 |
| | CORPSE NEP | -0.03 | 0.47 | 1571 |
| | MIMICS NEP | -0.09 | 0.62 | 732 |
| | | | | 0 |
| | CASA HR | 14.26 | 0.10 | 1 |
| 0°-23°S | CORPSE HR | 14.30 | 0.12 | 1 |
| | MIMICS HR | 14.24 | 0.29 | 2 |
| | CASA NPP | 14.52 | 0.57 | 4 |
| | CASA NEP | -0.27 | 0.61 | 232 |
| | CORPSE NEP | -0.22 | 0.58 | 262 |
| | MIMICS NEP | -0.28 | 0.84 | 296 |
| | | | | 0 |
| | CASA HR | 3.72 | 0.05 | 1 |
| 24°-90°N | CORPSE HR | 3.74 | 0.07 | 2 |
| | MIMICS HR | 3.74 | 0.08 | 2 |
| | CASA NPP | 3.77 | 0.26 | 7 |

| | | | |
|---|---|---|---|
| CASA NEP | -0.05 | 0.26 | 518 |
| CORPSE NEP | -0.03 | 0.24 | 810 |
| MIMICS NEP | -0.03 | 0.33 | 1135 |

Table S3: Multiple linear regression coefficients ($\gamma$) and $R^2$ are used to model interannual variability in heterotrophic respiration as a function of interannual variability in temperature,

NPP, or preceding year NPP. All variables have been detrended and deseasonalized. We list statistically significant predictors of HR IAV, as determined by p-values from ANOVA.

| Region | Model IAV | HR IAV Regression $\gamma^*$ , $R^2$ | | |
|---|---|---|---|---|
| | | CASA-CNP Temperature IAV [Pg C y$^{-1}$ K$^{-1}$] | CASA-CNP NPP Current year IAV [--] | CASA-CNP NPP Preceding year IAV [--] |
| 61°-90°N | CASA HR | **0.16** , 0.64 | **0.74** , 0.67 | -0.14, 0.02 |
| | CORPSE HR | **0.42**, 0.54 | **2.22**, 0.77 | -0.23, 0.01 |
| | MIMICS HR | **0.13**, 0.32 | **0.63**, 0.40 | 0.06, 0.00 |
| | CASA-CNP NPP | **0.15**, 0.43 | | |
| 24°-60°N | CASA HR | **0.78**, 0.58 | 0.2, 0.05 | 0.33, 0.15 |
| | CORPSE HR | **1.00**, 0.57 | -0.28, 0.06 | 0.32, 0.08 |
| | MIMICS HR | **1.74**, 0.70 | **-0.90**, 0.27 | 0.05, 0.00 |
| | CASA-CNP NPP | -0.38, 0.10 | | |
| 1°-23°N | CASA HR | 0.17, 0.14 | **-0.10**, 0.26 | **0.12**, 0.45 |
| | CORPSE HR | 0.00, 0.00 | -0.06, 0.07 | **0.17**, 0.61 |
| | MIMICS HR | **1.03**, 0.68 | **-0.40**, 0.60 | -0.03, 0.00 |
| | CASA-CNP NPP | **-1.57**, 0.42 | | |
| 0°-23°S | CASA HR | **0.24**, 0.17 | **-0.07**, 0.17 | **0.12**, 0.41 |
| | CORPSE HR | -0.01, 0.00 | -0.01, 0.00 | **0.17**, 0.61 |
| | MIMICS HR | **1.50**, 0.87 | **-0.46**, 0.79 | 0.03, 0.00 |
| | CASA-CNP NPP | **-2.65**, 0.72 | | |
| 24°-90°N | CASA | 0.00, 0.00 | 0.03, 0.03 | **0.12**, 0.32 |
| | CORPSE | -0.07, 0.04 | 0.09, 0.13 | **0.17**, 0.36 |
| | MIMICS | **0.32**, 0.42 | **-0.26**, 0.65 | -0.08, 0.04 |
| | CASA-CNP NPP | **-1.11**, 0.52 | | |

*bolded values are statistically significant (p < 0.05)

Figure S1: Depiction of interannual variability (IAV) calculation. (a) Multi-site mean CASA-CNP $CO_2^{NEP}$ in the Northern Hemisphere high latitudes (NHL) region for 1982 to 2010 ($CO_2^{NEP}$ = $CO_2^{HR}$ + $CO_2^{NPP}$). (b) Detrended CASA-CNP $CO_2^{NEP}$ timeseries after removing a third-order polynomit fit. (c) Climatological annual cycle calculated using the median of monthly values over the analysis period. (d) CASA-CNP $CO_2^{NEP}$ interannual variability calculated from removing the climatological annual cycle from each year in the detrended timeseries.

Figure S2: $CO_2$ concentrations from ocean fluxes. $CO_2$ output from GEOS Chem simulation forced by $CO_2$ CAMS inversion 17r1. (a) Averaged $CO_2$ in the Northern Hemisphere high latitudes (NHL) region for 1982 to 2010. (b) Detrended $CO_2$ timeseries after removing a third-order polynomit fit. (c) Climatological annual cycle calculated using the median of monthly values over the analysis period. (d) $CO_2$ interannual variability calculated from removing the climatological annual cycle from each year in the detrended timeseries.

Figure S3: $CO_2$ concentrations from fossil fuel emissions. $CO_2$ output from GEOS Chem simulation forced by $CO_2$ CAMS inversion 17r1. (a) Averaged $CO_2$ in the Northern Hemisphere high latitudes (NHL) region for 1982 to 2010. (b) Detrended $CO_2$ timeseries after removing a third-order polynomit fit. (c) Climatological annual cycle calculated using the median of monthly values over the analysis period. (d) $CO_2$ interannual variability calculated from removing the climatological annual cycle from each year in the detrended timeseries.

**Land mask Methodology: CASA Soil Heterotrophic Respiration (HR)**

Figure S4: Example land mask for the Northern Hemisphere High Latitudes (NHL). (a) CASA-CNP soil heterotrophic respiration (HR) for January 1982. (b) Global land mask with land gridcells set to a constant fill value and ocean gridcells set to NaN. (c) Land mask after application of a separate NHL latitude band mask with values of 0 south of 61 °N. Land gridcells were reassigned a value of 1. (d) NHL land mask weighted by area with Greenland removed.

---

## Author Comment (AC3) · 11 Jan 2020

Basile and colleagues compare three model formulations of heterotrophic respiration in their predictions of CO2 generation to the atmosphere, and compare the predictions with observations of atmospheric CO2 concentrations from a series of oceanic observations. I found the direct comparison of model formulations to be important and timely, given that the authors compared a CENTURY-like traditional formulation to more recent "mechanistic" models that explicitly simulate microbial processes. In some ways, this is a well-written manuscript. The text has the crisp precision is a hallmark of good scientific writing. However, the manuscript is also challenging to understand and follow,

in part because it uses jargon as well as many symbols and acronyms. I suggest that the authors embed summary sentences at the end of some paragraphs throughout the results and discussion section to sum up the meaning of the results for the reader, without using acronyms (e.g., "These results suggest that the preponderance of the CO2 production driving the seasonal cycle of atmospheric CO2 originates in the southern tropical region").

We thank the reviewer for their recognition of where this work falls in the modeling field and their feedback for more clear connecting statements and explanations throughout the paper. We have added summary sentences without jargon or abbreviations throughout the Discussion section of the revised manuscript.

Specific remarks: Line 60. I agree with the central message of this paragraph, but I would further emphasize that HR is exceptionally challenging to measure, even at the local scale. Separating soil respiration into autotrophic and heterotrophic components is possible, and it has been done well in a few places where isotopic techniques are possible on intact soils, but it has also been done poorly or with significant limitations in other places. This is, in part, because of the intrinsic linkage between microbial decomposition and root activity (i.e., exudation, allocation of carbohydrate to mycorrhizal partners). I encourage the authors to acknowledge the uncertainty in estimating HR from soil respiration fluxes, similar to their statement regarding NEE measurements (âĹijline 64).

The paragraph has been reworded as:

"Ecosystem respiration, or the combination of autotrophic and heterotrophic respiration fluxes, can be backed out from eddy covariance net ecosystem exchange observations at spatial scales around 1 km2, but with substantial uncertainty (Baldocchi 2008; Barba et al., 2018; Lavigne et al., 1997). The bulk of ecosystem respiration fluxes comes from soils, but soil respiration fluxes from chamber measurements can exceed ecosystem respiration measurements from flux towers, highlighting uncertainties in integrating

spatial and temporal variability in ecosystem and soil respiration measurements (Barba et al. 2018). Further partitioning of soil respiration measurements into autotrophic and heterotrophic components to derive their appropriate environmental sensitivities remains challenging, but critical to determining net ecosystem exchange of $CO_2$ with the atmosphere (Bond-Lamberty et al 2004, 2011, 2018). Additionally, because fine-scale variations in environmental drivers such as soil type and soil moisture affect rates of soil respiration, it is difficult to scale local respiration observations to regional or global levels (but see Zhao et al. 2017). Specifically, soil heterotrophic respiration (HR), the combination of litter decay and microbial breakdown of organic matter, is the main pathway for $CO_2$ release from soil carbon pools to the atmosphere. Currently, insights on HR rates and controls are mostly derived from local-scale observations. Soil chamber observations can be used to measure soil respiration at spatial scales on the order of 100 cm2 (Davidson et al., 2002; Pumpanen et al., 2004; Ryan and Law, 2005). "

Line 80. I appreciate this text directly comparing models like CENTURY to the newer, "more mechanistic" models that explicitly simulate microbial processes. Directly comparing these modeling frameworks is timely and important.

We thank the reviewer for this positive note and kept this comment in perspective with the feedback for additional model comparison.

Line 239. I am somewhat concerned by the lack of treatment of ocean $CO_2$ fluxes, which are quantitatively large relative to the other fluxes listed here. I appreciate the following sentence, which at least partially addresses my concern. The authors might consider specifically state the assumption they are making by ignore these fluxes, which is that ocean $CO_2$ fluxes are constant at seasonal and interannual timescales. This assumption is challenging to swallow, particularly given that the atmospheric $CO_2$ observations were made in areas surrounded by oceans.

We acknowledge the reviewer's concern on the assumption surrounding ocean fluxes. We have updated the text with the following statements and have added Supplementary Figures (SFig 1-2) that show the magnitude of ocean flux contributions to atmospheric $CO_2$ in comparison with CASA-CNP CO2NEP for the Northern Hemisphere high latitudes.

"We also simulated the fossil and ocean imprint on atmospheric $CO_2$ using boundary conditions from CO2 CAMS inversion 17r1 (https://atmosphere.copernicus.eu/sites/default/files/2018-10/CAMS73_2015SC3_D73.1.4.2-1979-2017-v1_201807_v1-1.pdf). However, at the temporal scales of this analysis, ocean and fossil fuel fluxes had a much smaller influence on regional patterns of atmospheric $CO_2$ than did land fluxes. Across the six latitude bands, the detrended CO2NEP annual amplitude ranges from a factor of 1.5 (in the tropics) to an order of magnitude larger (at high latitudes) than $CO_2$ from ocean fluxes and fossil fuel emissions. Likewise, the IAV from fossil and ocean-derived $CO_2$ was at most 25% that of NEP-derived $CO_2$ at most latitude bands. These results are consistent with previous studies that have demonstrated that NEP drives most of the atmospheric $CO_2$ seasonality (> 90%; Nevison et al., 2008; Randerson et al., 1997) and interannual variability (e.g., Rayner et al. 2008; Battel et al. 2000). Given that patterns of IAV in ocean and fossil $CO_2$ partially cancel each other and the large uncertainty in ocean fluxes, we choose to omit these $CO_2$ species from our analysis."

I found the ordering of the results to be challenging to understand. I first wanted to see an assessment of the model simulations relative to the data at the two temporal scales of interest here (seasonal and interannual).

We find this point helpful to improve the readability of the paper. We have restructured the seasonality text in the Results and Discussion sections to further distinguish HR impacts. The following text was moved from the Discussion section 4.1 to section 3.1:

"Our evaluation of $CO_2$ simulated using testbed fluxes revealed that all testbed models overestimated the mean annual cycle amplitude of atmospheric $CO_2$ observations. In the Northern Hemisphere, the bias was largest for MIMICS, as the CO2MIMICS NEP

amplitude was overestimated by up to 100% (Fig. 3). The mismatch was smallest in CO2CORPSE NEP, which was within 70% of the observed annual cycle amplitude where CORPSE simulates the largest seasonal HR fluxes (Fig. 3a-c, Table 1)."

The following text was added to the Discussion under section 4.1:

"Our evaluation of $CO_2$ simulated using testbed fluxes revealed that all testbed models overestimated the mean annual cycle amplitude of atmospheric $CO_2$ observations. In the Northern Hemisphere, the bias was largest for MIMICS, which had a $CO_2$ amplitude from net ecosystem production that was overestimated by up to 100% (Fig. 3). The mismatch in the amplitude of the Northern Hemisphere NEP fluxes was smallest from CORPSE, despite CORPSE also simulating the largest seasonal amplitude in HR fluxes (Fig. 3a-c, Table1). By contrast, in the Southern Hemisphere the simulated $CO_2$ annual cycle amplitudes were similar across all three models, with small absolute mismatches (about 1 ppm) compared to observations (Fig. 3)."

I did not find Figure 2 or it's associated text at the beginning of the results section to be useful in aiding my understanding. I am likely missing something. However, I would find the results to be structured more understandably (for me) if the current figures 3 and 4 became the first figures presented as results. That is, the authors may consider omitting figure 2, or moving it down.

We've spent some time clarifying the results associated with Fig. 2, now Fig. 4 after rearranging. But because the integrated effects of the difference between NPP and heterotrophic respiration (Fig. 4) are reflected in the $CO_2$ NEP fluxes (Fig. 3), we still see value in presenting the component fluxes.

I was surprised by the relative lack of direct comparisons across these three models in the discussion section. I was hoping for more explicit "unpacking" of the particular model formulations, with direct recommendations as to which model components are most justifiable given the observed data. I found that much of the discussion amounted to throw-away sentences such as line 457-460, in which little of consequence was said

regarding how we should model HR.

We thank the reviewer for this comment to clarify our discussion points and add more detail for comparison between the testbed models. The discussion was extensively modified to add depth to the comparisons being made and avoid jargon and abbreviations. We also added clarifying sentences on the scope of the analysis in the introduction section.

Please also note the supplement to this comment:
https://www.biogeosciences-discuss.net/bg-2019-256/bg-2019-256-AC3-supplement.pdf

—————————————————————————

[Figure]

**Fig. 1.** Figure 1: Flow chart depiction of the analysis process from soil model fluxes to simulated CO2 concentration and comparison with NOAA observations.

[Figure]

**Fig. 2.** Figure 2: Tagged flux regions and marine boundary layer CO2 observing sites used in our analysis. The 5 tagged flux regions are shown in color fill: Northern High Latitude (NHL), Northern Mid-Latitude

[Figure]

**Fig. 3.** Figure 3: Climatological annual cycle (median) of CO2 for observations (black) and global net ecosystem productivity flux (CO2NEP, colors) between 1982 and 2010. Monthly climatology values were create

[Figure]

(a) 61–90°N
(b) 24–60°N
(c) 1–23°N
(d) 0–23°S
(e) 24–60°S
(f) 61–90°S

Legend: ESRL, CASA–CNP, CORPSE, MIMICS

**Fig. 4.** Figure 4: Climatological annual cycle (median) of atmospheric CO2 simulated from land fluxes (CO2NPP, CO2HR) between 1982 and 2010. Monthly climatology values were created after detrending the CO2 tim

[Figure]

Fig. 5. Figure 5: Interannual variability of CO2 from global net ecosystem productivity (CO2NEP IAV) for testbed models (colors) and marine boundary layer observations from the NOAA ESRL network (black). Gra

[Figure]

[Figure]

**Fig. 6.** Figure 6: Magnitude of CO2 interannual variability resulting from (a) individual flux components (CO2NPP IAV, CO2HR IAV) and (b) global net ecosystem productivity (CO2NEP IAV). Observed CO2 IAV from N

[Figure]

**Fig. 7.** Figure 7: Temperature sensitivity ($\gamma$) calculated for interannual variability (IAV) of CASA-CNP air temperature and (a) flux IAV and corresponding CO2 growth rate anomalies, (b) NEP IAV and CO2NEP grow

[Figure]

**Fig. 8.** Figure 8: Comparison of regional and global interannual variability (IAV) from land fluxes and resulting atmospheric CO2 between 1982 and 2010. (a, c, e) Normalized ratio taken between regional IAV an

---

## Author Response (AR1)

**This study uses globally modeled ecosystem CO₂ fluxes and an atmospheric transport model together with observed atmospheric CO₂ concentrations to evaluate different soil respiration models. This is done by comparing the inter-annual and intra-annual variability in CO₂ fluxes between model and observations. The global scale of the study makes this a rather rough comparison, which however is useful for identifying soil models that perform better at such scales. In general, there seems to be a considerable amount of work and careful analysis carried out and the study had scientifically interesting results. The manuscript has good grammar and few typos. The introduction goes from the more general to the specific aims and summary of the methods and is well written. On the down side, much in the methods and results remains difficult to follow. Explaining the procedure in more detail will largely facilitate understanding the results and discussion. Generally, the text should be made more readable to a biogeochemistry audience that may not be familiar with the jargon and assumptions used in this specific field. Because there are many variables and procedures being used, it would help to more carefully define each part and consistently use the defined terms throughout the text. The manuscript should reach publication quality after these aspects are taken care of.**

We thank the reviewer for their time and comments. We appreciate there are elements of the paper that need more translation between the atmospheric and biogeochemical communities. In response to the reviewer's suggestions, we have clarified definitions and explanations throughout the paper. We will explicitly include the following text and equations in sections 2.1 and 2.3. We further address the reviewer's individual comments below.

Text and equations added to section 2.2:

*"The testbed is a chain of model simulations where soil models with different structures can be run under the same forcing data, including the same gross primary productivity (GPP) fluxes, soil temperature, and soil moisture. The testbed produces its own estimates of net primary production (NPP), the difference between GPP and autotrophic respiration (AR; Eqn. 1). Each testbed soil model in this analysis produces unique gridded heterotrophic respiration (HR) values based on its own underlying mechanism and soil C stocks.*
*...*
*From the testbed output we calculate the net ecosystem productivity (NEP; Eqn. 3). In the analysis presented here, CASA NPP was used across the testbed ensemble in the NEP calculation, thus highlighting differences in the timing and magnitude of HR fluxes from the individual soil models. From a land perspective (positive NEP fluxes into land), NEP is calculated as NPP – HR, where respiration release of CO₂ decreases net carbon gains through photosynthesis. Here, we use an atmospheric perspective for NEP (positive NEP fluxes into the atmosphere) by reversing the sign on the NPP flux and taking HR as positive (Eqn. 3)."*

$$NPP = GPP - AR \qquad (1)$$
$$NEP = HR - NPP \qquad (2)$$

*...*

Text and equations added to section 2.3:

*"Throughout the manuscript, we refer to $CO_2$ originating from these NPP and HR component fluxes as $CO_2{}^{NPP}$ and $CO_2{}^{HR}$, respectively. We use a sign convention for the fluxes whereby a positive value indicates a source of carbon to the atmosphere, which means we can combine the $CO_2$ tracers from NPP and HR to calculate the expected atmospheric variation owing to NEP using (Eqn. 3):*

$$CO_2^{NEP} = CO_2^{HR} + CO_2^{NPP} \qquad (3)$$

**Specific remarks:**

**L44: is this not rather a "spatial" rather than "concentration" footprint?**

We thank the reviewer for this comment, as concentration footprint is an atmospheric science term that may be less familiar to the biogeochemistry community. We have therefore changed the usage in the revised manuscript for Lines 42-44:

*"The high precision and accuracy of in situ observations and the fact that these measurements integrate information about ecosystem carbon fluxes over a large spatial footprint make atmospheric $CO_2$ a strong constraint on model predictions of net carbon exchange (Keppel-Aleks et al., 2013)."*

**L108: The term "phasing" is used in several places. Please describe its meaning on first use.**

The term was updated in this line as the following:

*"For example, at seasonal timescales, improving the parameterization for litterfall in the CASA model improved phasing – i.e., the timing of seasonal maxima, minima, and inflection points – for the simulated atmospheric $CO_2$ annual cycle (Randerson et al., 1996)."*

**L114: remove "were"**

The sentence was updated.

**L133-134: Please explain in more detail what you mean by diagnosing using one standard deviation.**

We have modified the text at Lines 148-150 to describe the use of standard deviation to calculate the magnitude of interannual variability:

*"The magnitude of $CO_2$ IAV is calculated as one standard deviation on the detrended, deseasonalized timeseries, unless otherwise noted."*

**L138-142: Six latitude zones are mentioned but 5 used later. Maybe clarify here that 2 zones are aggregated. Also, could you use the same naming for north and south (e.g. NHL, SHL, etc.).**

The distinction between latitude bands used for aggregating atmospheric $CO_2$ observations and the latitude bands used for the flux analysis is made in the revised manuscript at Lines 139 to 142. The hemisphere naming convention has been updated at Lines 140 to 142 to reflect the same abbreviations used later in the text:

*"Northern Hemisphere high latitudes (NHL: 61 to 90°N), midlatitudes (NML: 24 to 60°N), tropics (NT: 1 to 23°N), Southern Hemisphere tropics (ST: 0 to 23°S), and two Southern extratropics bands: the southern midlatitudes (SML, 24-60 S), and the southern high latitudes (SHL, 61-90 S). The global mean $CO_2$ timeseries is constructed as an area-weighted average of these six atmospheric zones."*

**L162-166: CLM can be run to simulate the mentioned fluxes. Why was CASA used instead?**

The idea behind the testbed was to generate a more generalizable framework for model comparisons that could take observations (point scale simulation forced with flux tower data) or modeled output from land models like CLM, ELM, DAYCENT, etc and deliver consistent forcings to the soil models (CASA, MIMICS, CORPSE). This would allow us also to look at forcing uncertainty from other models (which is not the focus of the work we presented). For these runs we chose to use GPP, soil moisture, soil temperature, etc from CLM simulations (CLM4.5-SP with Cru-NCEP forcing) as the input for the testbed, in part because this model configuration has been documented in Wieder et al., 2018. For the analysis presented in this paper, CASA NPP was used across the soil model ensemble in the NEP calculation for consistency in highlighting HR differences.

The following sentence was added to the text for clarification in section 2.2 at Lines 176-179:

*"In the analysis presented here, CASA NPP was used across the testbed ensemble in the NEP calculation, thus highlighting differences in the timing and magnitude of HR fluxes from the individual soil models."*

**L194: model structure?**

We provide a definition of model structure in the updated text at Lines 209-213:

*"Overall, while the testbed approach contains necessary simplifications, it provides the ability to query the role of model structure, including assumptions about the number of soil carbon pools, the role of microorganisms, and the sensitivity to environmental factors, in driving HR flux differences when NPP and environmental controls are held in common."*

**L196-197: The procedure here is unclear. What do you mean by masked to land regions that align with the sampling zones? Do you mean the latitude zones defined earlier? Why do you specify "land regions", is it because there are ocean fluxes that need to be excluded? "mask" is a term that many reader will not understand. Why are monthly averages calculated if daily values are used for the transport model? Please add detail.**

We have used the reviewer's comment as an opportunity to streamline the methods, and we have also added a flow diagram based on Reviewer 2's comment. The text has been updated as follows for Lines 215-219:

*"The testbed fluxes are used in two ways: first, we analyze monthly-averaged, regional fluxes for net primary production (NPP) from CASA-CNP and HR simulated by CASA-CNP, CORPSE and MIMICS. Second, we use the raw daily fluxes as boundary conditions for global GEOS-Chem runs to simulate the influence of these fluxes on atmospheric $CO_2$, as described in the following section."*

**L197-199: A repetition of the zones does not seem necessary.**

This has been removed.

**L203-204: Please give more detail, either here or below, of how exactly these are used as boundary conditions for the model.**

Based on the reviewer's comment, we modified the order of this description to section 2.3 and provided the following details at Lines 228-239:

*"The testbed fluxes from 1980 to 2010 are used for land emissions to simulate the imprint of these different soil model configurations on atmospheric $CO_2$ (Fig. 1). In our simulations, HR and NPP fluxes were separated into the five regions listed above (NHL, NML, NT, ST, SE) so that the influence of carbon fluxes originating from these individual regions on global atmospheric $CO_2$ mole fraction could be quantified. We initialized separate species of $CO_2$ in the atmospheric model, one for each flux (HR or NPP) and region (NHL, NML, etc.). Since we considered four fluxes (CASA NPP and three types of HR) originating in five regions, we simulated a total of 20 species. These species were tracked throughout the simulation as their spatiotemporal distribution changed due to the combined influence of $CO_2$ fluxes at the surface and atmospheric weather. Although these species are simulated individually, we can simply sum the regional atmospheric species for a given flux (e.g., CASA HR) to determine the atmospheric $CO_2$ arising from all fluxes over the globe."*

**L204: Do you mean the transport model is run for lat zones separately? How does is deal with the lat boundaries in this case?**

Simulations are all run globally, so there is no issue with latitudinal boundary conditions. The text has been updated in the previous comment to explain the use of land masking to isolated latitudes from each other in the emissions component of the model simulation.

**L211: "data" rather than "fields"**

This change was made in the revised manuscript.

**L215: Explain here the model CO₂ flux inputs and outputs (presumably the boundary conditions mentioned before?) and clarify what fluxes are omitted (fossil fuels, etc)**

We updated the text at Lines 239-251 to reflect that we did simulate the contribution of ocean and fossil $CO_2$, but neglect the $CO_2$ mole fraction owing to these fluxes:

> *"We also simulated the fossil and ocean imprint on atmospheric $CO_2$ using boundary conditions from $CO_2$ CAMS inversion 17r1 (https://atmosphere.copernicus.eu/sites/default/files/2018-10/CAMS73_2015SC3_D73.1.4.2-1979-2017-v1_201807_v1-1.pdf). However, at the temporal scales of this analysis, ocean and fossil fuel fluxes had a much smaller influence on regional patterns of atmospheric $CO_2$ than did land fluxes. Across the six latitude bands, the detrended $CO_2^{NEP}$ annual amplitude ranges from a factor of 1.5 (in the tropics) to an order of magnitude larger (at high latitudes) than $CO_2$ from ocean fluxes and fossil fuel emissions. Likewise, the IAV from fossil and ocean-derived $CO_2$ was at most 25% that of NEP-derived $CO_2$ at most latitude bands. These results are consistent with previous studies that have demonstrated that NEP drives most of the atmospheric $CO_2$ seasonality (> 90%; Nevison et al., 2008; Randerson et al., 1997) and interannual variability (e.g., Rayner et al. 2008; Battle et al. 2000). Given that patterns of IAV in ocean and fossil $CO_2$ partially cancel each other and the large uncertainty in ocean fluxes, we choose to omit these $CO_2$ species from our analysis."*

**L216: "minimize the"**

The text has been updated following this and the next comment.

**L217: Confusing. Anthropogenic emissions are not modeled, so they cannot have an influence. Presumably you are relating this to the observations. Please reword.**

The reviewer is correct that this was done to improve the utility of atmospheric observations as a benchmark for the simulations, and we have revised the text to read as follows at Lines 253-262:

> *"We sample the gridded atmospheric simulation output at the 34 marine boundary layer (MBL) sites identified in section 2.1, using the 3ʳᵈ vertical level to minimize influence of land-atmosphere boundary layer dynamics. We then calculate the latitude zone average, median annual cycle and interannual variability using the methods described for $CO_2$ observations (see section 2.1). Averaging $CO_2$ from all sites within a latitude band is consistent with our hypothesis that atmospheric $CO_2$ may provide constraints on large-scale patterns of heterotrophic respiration, but individual sites may be too heavily influenced by local characteristics not accounted for by the testbed fluxes. As such, averaging simulated and observed $CO_2$ across latitude zones smooths local information while retaining information about regional scale fluxes."*

**L219: "closest to the observation sites, i.e. the"**

Wording correction was addressed in response to the previous comment.

**L221: remove "calculations"**

Text was removed.

**L222-225: This clarification should come before, when the averaging is explained. Although, as stated, atmospheric [CO$_2$] over the ocean should integrate the signal over large regions, it is not clear why the zones and averaging was necessary. Since model data can be obtained for the observation points, the residuals of these could have been analyzed directly.**

Given the generalized framework of the testbed, and the fact that it doesn't account for highly local processes that might affect the CO$_2$ observations, we prefer to use the zonal averaging approach we outlined rather than compare with individual sites. However, we have added clarification as the reviewer suggested at Lines 257-262:

> *"Averaging CO$_2$ from all sites within a latitude band is consistent with our hypothesis that atmospheric CO$_2$ may provide constraints on large-scale patterns of heterotrophic respiration, but individual sites may be too heavily influenced by local characteristics not accounted for by the testbed fluxes. As such, averaging simulated and observed CO$_2$ across latitude zones smooths local information while retaining information about regional scale fluxes."*

**L226-242: This section is not clear.**

Text changes in section 2.2 have expanded the explanation of NPP and HR as boundary conditions for the GEOS Chem atmospheric transport model simulations. A figure demonstrating the land masking technique used has been added to the Supplementary information as well as a chart detailing the input fluxes and resulting CO$_2$ naming convention. Additionally, Line 242 has been moved to section 2.2 to explain the use of NEP as a variable with the accompanying equation defining the term.

**L226: "spatial" you mean the lat zones? How exactly do you isolate the imprints? What do you mean by "tag"? I believe "track CO$_2$ tracers" is confusing since "tracer" relates to flow dynamics, but the analysis is about concentration changes(?). You mention "4 sets of fluxes" but aren't the observations CO$_2$ concentrations. What is then being compared (you mention CO$_2^{NEP}$ variability is comparable to observations)? Are CO$_2^{NEP}$ and CO$_2^{HR}$ fluxes or concentrations? If these are the fluxes, should they not be introduced in the previous section?**

We thank the reviewer for this reminder to remove atmospheric science jargon from the paper. We have replaced the use of the word "tracer" with "species" throughout to indicate that we consider each region to affect only one CO$_2$ species. The concentration of each species is then directly tied back to a single flux. The updated text at Lines 228-239 reads as the following:

> *"The testbed fluxes from 1980 to 2010 are used for land emissions to simulate the imprint of these different soil model configurations on atmospheric CO$_2$ (Fig. 1). In our simulations, HR and NPP fluxes were separated into the five regions listed above (NHL, NML, NT, ST, SE) so*

*that the influence of carbon fluxes originating from these individual regions on global atmospheric $CO_2$ mole fraction could be quantified. We initialized separate species of $CO_2$ in the atmospheric model, one for each flux (HR or NPP) and region (NHL, NML, etc.). Since we considered four fluxes (CASA NPP and three types of HR) originating in five regions, we simulated a total of 20 species. These species were tracked throughout the simulation as their spatiotemporal distribution changed due to the combined influence of $CO_2$ fluxes at the surface and atmospheric weather. Although these species are simulated individually, we can simply sum the regional atmospheric species for a given flux (e.g., CASA HR) to determine the atmospheric $CO_2$ arising from all fluxes over the globe."*

All modeled $CO_2$ is output as concentrations (ppm) and the ESRL $CO_2$ observations are also reported as mole fraction (ppm). Additional wording was added to section 2 to clarify variable units:

Lines 133 to 135:

*"For this analysis we use reference $CO_2$ measurements reported in parts per million (ppm) from 34 marine boundary layer sites (MBL, Table S1) within the NOAA Earth System Research Laboratory sampling network (ESRL, Fig. 1; Dlugokencky et al., 2016)."*

Lines 156 to 157:

*"All testbed fluxes are output in grams of carbon per meter square at a daily temporal resolution and then converted to petagrams (Pg C, over a region)."*

Lines 227 to 228:

*"The model is initialized with globally-uniform atmospheric $CO_2$ mole fraction equal to 350 ppm."*

**L245-248: The calculation is not the variability but just the growth rate (or rate of change). Where does the annual IAV time series come from? Is it an annual average of the IAV series? Please clarify. Consider rewording, e.g. "The monthly and yearly rate of change of the the interannual anomaly (i.e. the IAV timeseries) was calculated as ...".**

The text was reworded at Lines 279-286 to explain the calculation of the growth rate anomaly (rate of change in the interannual $CO_2$ values, or residual values after detrending and removing the seasonal cycle) which is a distinct value from the growth rate (rate of change in the unaltered $CO_2$ values):

*"Rates of change were derived from monthly and annual timeseries to ultimately calculate the temperature sensitivity of the testbed fluxes, the modeled $CO_2$, and the observed $CO_2$ values. The $CO_2$ growth rate anomaly was calculated as the difference between timestep n and n-1 in both the monthly and annual $CO_2$ IAV timeseries. As a result of this technique, the monthly $CO_2$ growth rate anomalies were centered on the first day of the corresponding months. To compare flux information with $CO_2$ growth rate anomalies, daily testbed flux timeseries were*

*averaged to monthly resolution and then interpolated by averaging between months to center values on the first day of each month."*

**L248-250: This sentence is not understandable.**

This wording was replaced in the response to the previous comment.

**L251-253: "fitted to ... against ..." instead of "for .. with ...". Also, where does T IAV come from?**

The text has been reworded at Lines 287-290 as:

*"Following Arora et al. (2013), we calculate temperature sensitivity (γ) using an ordinary linear regression (OLR). We calculate OLR for the interannual variability timeseries of CASA-CNP soil temperature (T IAV) against 1) atmospheric $CO_2$ growth rate anomalies, and 2) land flux IAV (see section 2.2)."*

**L255: "global temperature sensitivity"**

The text has been updated.

**L255-259: Was this reference calculated from IAV data or actual observations? What is this reference for?**

The term "reference" was used needlessly in the original manuscript, and the text has been updated at Lines 292-295 to the following:

*"The global temperature sensitivity value for the observed $CO_2$ growth rate anomaly was calculated for 1982 to 2010 using ESRL $CO_2$ observations and the Climatic Research Unit's gridded temperature product (CRU TS4; Jones et al., 2012), which is derived from interpolated ground station measurements."*

**L263-277: Global variability of what? What do you define as SDrel, the ratio or the "IAV magnitude"? (presumably the latter, but be more explicit). In any case, the term IAV magnitude is confusing. Why not simply say IAV relative SD? The calculation should be made clearer. What are "regional values of simulated $CO_2$" (since all simulations made are regional $CO_2$). Why do you use the relative standard deviation (or CV). A high CV from a small flux can have a smaller impact on global values than a small CV from a large flux. The actual SD may be a better measure. Also, it seems to me that using a ratio between regions and global assumes an additive effect of each region, which might not be the case (what if the regional IAV CV is larger than the global value?). What do you mean by sourced only from a single region? Consider rewriting this paragraph/revising this analysis.**

We thank the reviewer for this comment and the opportunity to clarify the methods we used. For the flux analysis, the SDrel is the ratio of the IAV magnitude in a given region relative to the IAV magnitude of the globe (five regions summed). For the $CO_2$ analysis, the SDrel is the ratio of the IAV magnitude of the global-average atmospheric $CO_2$ arising from a single region relative to the IAV magnitude from the global fluxes. Thus, we are not using a coefficient of

variability, which would normalize the magnitude of IAV to the baseline flux. As the reviewer notes, we avoid this formulation because it could lead to a high CV when fluxes are small, which is undesirable. We agree with the reviewer that the SDrel from the five regions is not necessarily additive, which is why we have provided correlation coefficients between the regional and global information since not all variations will be coherent with the global signal.  The text has been made clearer at Lines 296-312 as to why the values were chosen and how they are defined:

*"We also assess the influence of individual regions on the global mean signal for both component land fluxes (NPP, HR) and simulated atmospheric $CO_2$ ($CO_2^{NPP}$, $CO_2^{HR}$, $CO_2^{NEP}$). We first quantify the magnitude of variability in each region relative to the magnitude of global variability ($\sigma_{REL}$) as the ratio of regional IAV standard deviation to global IAV standard deviation. This ratio is calculated for monthly flux IAV from each of the five flux regions and for the global-mean $CO_2$ timeseries that arises from fluxes in each of the five flux regions (e.g., the global $CO_2$ response to NHL fluxes, or the global $CO_2$ response to NML fluxes, etc.). The value of $\sigma_{REL}$ has a lower bound of 0, which would indicate that a region contributes no IAV, but has no upper bound, since a value greater than 1 simply indicates that the fluxes in a given region are more variable than global fluxes.*

*We note that the timing of IAV in a given region may be independent of IAV in other regions, and thus may or may not be temporally in-phase with global IAV. We therefore also calculate correlation coefficients (r) for the timeseries of regional flux IAV and $CO_2$ IAV with the global signal. Thus, if an individual region were responsible for all observed global flux or $CO_2$ variability, it would have both $\sigma_{REL}$ and r values equal to 1 in this comparison. The value for r will be small if a regional signal is not temporally coherent with the global signal, even if the magnitude of variability is high."*

**L283-285: Still at a loss of what $CO_2^{NPP}$, $CO_2^{HR}$ are. Atmospheric values derived from...? Is it in any way logical that peaks in $CO_2^{NPP}$ are max in April (early spring) and min August (mid summer) in NHL?**

Definitions for $CO_2^{NPP}$, $CO_2^{HR}$ are included in section 2.3. Also the peaks relate to the reversal in the sign of the NPP flux, which carries through to $CO_2^{NPP}$. Descriptive wording explaining the sign convention have been added to the text and Figure 2 (now Figure 4) caption.

The text at Lines 321-325 has been updated as:

*"Within the modeled carbon dioxide concentrations resulting from land fluxes, $CO_2^{NPP}$ and $CO_2^{HR}$, show largest seasonality in the NHL, with seasonal amplitudes decaying toward the tropics and Southern Hemisphere. In the NHL, the peak-to-trough amplitude of $CO_2^{NPP}$ is 39±2 ppm, with a seasonal maximum in April and a seasonal minimum in August (Fig. 4a; note, this $CO_2^{NPP}$ peak reflects the sign reversal in the driving NPP flux (section 2.3)."*

**L299-300: Does $CO_2^{NEP}$ not always consider NPP and HR together?**

The reviewer is correct that $CO_2^{NEP}$ always considers NPP and HR together, and equations have been added to section 2 to make this relationship clear.

**L338: Not if NPP increases more than HR.**

The text has been updated at Lines 372-375 for clarification:

> *"Both global NPP and HR fluxes are sensitive to temperature variations at interannual timescales, with increased build-up of $CO_2$ in the atmosphere at higher temperatures, in part because the rate of HR increases at higher temperature and in part because most latitude bands show a reduction in NPP at above-average temperatures."*

**Figure 2 and 3: If I understood correctly, values here are differences with the long term trend. To make this clearer, it would be good to note this in the caption.**

Clarification that the seasonal cycles shown are detrended values has been added to the figure captions.
**This paper connected heterotrophic respiration and atmospheric $CO_2$ concentrations, and examined impact of signals of heterotrophic respiration on atmospheric $CO_2$ using model outputs. This study suggests that the atmospheric $CO_2$ concentrations could be used to verify the global estimation of heterotrophic respiration, and furthermore verify models. The idea is interesting and the manuscript is basically clear. I have just a couple of comments.**

We thank the reviewer for their time and comments.

**The diagram of your analysis (kind of flowchart) is necessary to understand this analysis.**

We appreciate the need for clarification in this interdisciplinary work and have added the following chart to the figure list, along with an introductory description in section 2.

[Figure]

Figure 1: Flow chart depiction of the analysis process from soil model fluxes to simulated $CO_2$ concentration and comparison with NOAA observations.

**Seasonal cycles of HR in Fig. 2: The peaks of HR in middle and high latitudes of northern hemisphere are in autumn (Sep to Oct). Is this widely accepted? For me, the peaks should be in July-Sep. Are the heterotrophic respiration and NPP you used well correlated with observation data oriented estimates in terms of amount, seasonality, and spatial pattern? This is important for this study.**

The HR fluxes have been evaluated in Wieder et al., 2018. Figure 2 (now Figure 4) shows the seasonal cycle of atmospheric $CO_2$ arising from HR fluxes, thus showing the cumulative imprint of HR fluxes. While the HR flux might peak in July-Sep, the $CO_2$ concentration from HR continues to accumulate in the atmosphere leading to the peak in autumn.

**There seems several jargons which the authors should explain more carefully or replace them with easier wordings.**

We thank the reviewer for this note and have made a conscientious effort throughout the revisions to add additional explanation and remove jargon.
**Basile and colleagues compare three model formulations of heterotrophic respiration in their predictions of CO₂ generation to the atmosphere, and compare the predictions with observations of atmospheric CO₂ concentrations from a series of oceanic observations. I found the direct comparison of model formulations to be important and timely, given that the authors compared a CENTURY-like traditional formulation to more recent "mechanistic" models that explicitly simulate microbial processes. In some ways, this is a well-written manuscript. The text has the crisp precision is a hallmark of good scientific writing. However, the manuscript is also challenging to understand and follow, in part because it uses jargon as well as many symbols and acronyms. I suggest that the authors embed summary sentences at the end of some paragraphs throughout the results and discussion section to sum up the meaning of the results for the reader, without using acronyms (e.g., "These results suggest that the preponderance of the CO₂ production driving the seasonal cycle of atmospheric CO₂ originates in the southern tropical region").**

We thank the reviewer for their recognition of where this work falls in the modeling field and their feedback for more clear connecting statements and explanations throughout the paper. We have added summary sentences without jargon or abbreviations throughout the Discussion section of the revised manuscript.

**Specific remarks:**

**Line 60. I agree with the central message of this paragraph, but I would further emphasize that HR is exceptionally challenging to measure, even at the local scale. Separating soil respiration into autotrophic and heterotrophic components is possible, and it has been done well in a few places where isotopic techniques are possible on intact soils, but it has also been done poorly or with significant limitations in other places. This is, in part, because of the intrinsic linkage between microbial decomposition and root activity (i.e., exudation, allocation of carbohydrate to mycorrhizal partners). I encourage the authors to acknowledge the uncertainty in estimating HR from soil respiration fluxes, similar to their statement regarding NEE measurements (~line 64).**

The paragraph has been reworded at Lines 57-74 as:

*"Ecosystem respiration, or the combination of autotrophic and heterotrophic respiration fluxes, can be backed out from eddy covariance net ecosystem exchange observations at spatial scales around 1 km², but with substantial uncertainty (Baldocchi 2008; Barba et al., 2018; Lavigne et al., 1997). The bulk of ecosystem respiration fluxes comes from soils, but soil respiration fluxes from chamber measurements can exceed ecosystem respiration measurements*

*from flux towers, highlighting uncertainties in integrating spatial and temporal variability in ecosystem and soil respiration measurements (Barba et al. 2018). Further partitioning of soil respiration measurements into autotrophic and heterotrophic components to derive their appropriate environmental sensitivities remains challenging, but critical to determining net ecosystem exchange of $CO_2$ with the atmosphere (Bond-Lamberty et al 2004, 2011, 2018). Additionally, because fine-scale variations in environmental drivers such as soil type and soil moisture affect rates of soil respiration, it is difficult to scale local respiration observations to regional or global levels (but see Zhao et al. 2017). Specifically, soil heterotrophic respiration (HR), the combination of litter decay and microbial breakdown of organic matter, is the main pathway for $CO_2$ release from soil carbon pools to the atmosphere. Currently, insights on HR rates and controls are mostly derived from local-scale observations. Soil chamber observations can be used to measure soil respiration at spatial scales on the order of 100 $cm^2$ (Davidson et al., 2002; Pumpanen et al., 2004; Ryan and Law, 2005)."*

**Line 80. I appreciate this text directly comparing models like CENTURY to the newer, "more mechanistic" models that explicitly simulate microbial processes. Directly comparing these modeling frameworks is timely and important.**

We thank the reviewer for this positive note and kept this comment in perspective with the feedback for additional model comparison.

**Line 239. I am somewhat concerned by the lack of treatment of ocean $CO_2$ fluxes, which are quantitatively large relative to the other fluxes listed here. I appreciate the following sentence, which at least partially addresses my concern. The authors might consider specifically state the assumption they are making by ignore these fluxes, which is that ocean $CO_2$ fluxes are constant at seasonal and interannual timescales. This assumption is challenging to swallow, particularly given that the atmospheric $CO_2$ observations were made in areas surrounded by oceans.**

We acknowledge the reviewer's concern on the assumption surrounding ocean fluxes. We have updated the text with the following statements at Lines 239-251 and have added Supplementary Figures (SFig 1-2) that show the magnitude of ocean flux contributions to atmospheric $CO_2$ in comparison with CASA-CNP $CO_2^{NEP}$ for the Northern Hemisphere high latitudes.

*"We also simulated the fossil and ocean imprint on atmospheric $CO_2$ using boundary conditions from $CO_2$ CAMS inversion 17r1 ([https://atmosphere.copernicus.eu/sites/default/files/2018-10/CAMS73_2015SC3_D73.1.4.2-1979-2017-v1_201807_v1-1.pdf)](https://atmosphere.copernicus.eu/sites/default/files/2018-10/CAMS73_2015SC3_D73.1.4.2-1979-2017-v1_201807_v1-1.pdf)). However, at the temporal scales of this analysis, ocean and fossil fuel fluxes had a much smaller influence on regional patterns of atmospheric $CO_2$ than did land fluxes. Across the six latitude bands, the detrended $CO_2^{NEP}$ annual amplitude ranges from a factor of 1.5 (in the tropics) to an order of magnitude larger (at high latitudes) than $CO_2$ from ocean fluxes and fossil fuel emissions. Likewise, the IAV from fossil and ocean-derived $CO_2$ was*

*at most 25% that of NEP-derived CO$_2$ at most latitude bands. These results are consistent with previous studies that have demonstrated that NEP drives most of the atmospheric CO$_2$ seasonality (> 90%; Nevison et al., 2008; Randerson et al., 1997) and interannual variability (e.g., Rayner et al. 2008; Battel et al. 2000). Given that patterns of IAV in ocean and fossil CO$_2$ partially cancel each other and the large uncertainty in ocean fluxes, we choose to omit these CO$_2$ species from our analysis."*

**I found the ordering of the results to be challenging to understand. I first wanted to see an assessment of the model simulations relative to the data at the two temporal scales of interest here (seasonal and interannual).**

We find this point helpful to improve the readability of the paper. We have restructured the seasonality text in the Results and Discussion sections to further distinguish HR impacts.

The following text was moved from the Discussion section 4.1 to section 3.1:

*"Our evaluation of CO$_2$ simulated using testbed fluxes revealed that all testbed models overestimated the mean annual cycle amplitude of atmospheric CO$_2$ observations. In the Northern Hemisphere, the bias was largest for MIMICS, as the CO$_2^{MIMICS\ NEP}$ amplitude was overestimated by up to 100% (Fig. 3). The mismatch was smallest in CO$_2^{CORPSE\ NEP}$, which was within 70% of the observed annual cycle amplitude where CORPSE simulates the largest seasonal HR fluxes (Fig. 3a-c, Table 1)."*

The following text was added to the Discussion under section 4.1:

*"Our evaluation of CO$_2$ simulated using testbed fluxes revealed that all testbed models overestimated the mean annual cycle amplitude of atmospheric CO$_2$ observations. In the Northern Hemisphere, the bias was largest for MIMICS, which had a CO$_2$ amplitude from net ecosystem production that was overestimated by up to 100% (Fig. 3). The mismatch in the amplitude of the Northern Hemisphere NEP fluxes was smallest from CORPSE, despite CORPSE also simulating the largest seasonal amplitude in HR fluxes (Fig. 3a-c, Table 1). By contrast, in the Southern Hemisphere the simulated CO$_2$ annual cycle amplitudes were similar across all three models, with small absolute mismatches (about 1 ppm) compared to observations (Fig. 3)."*

**I did not find Figure 2 or it's associated text at the beginning of the results section to be useful in aiding my understanding. I am likely missing something. However, I would find the results to be structured more understandably (for me) if the current figures 3 and 4 became the first figures presented as results. That is, the authors may consider omitting figure 2, or moving it down.**

We've spent some time clarifying the results associated with Fig. 2, now Fig. 4 after rearranging. But because the integrated effects of the difference between NPP and heterotrophic respiration (Fig. 4) are reflected in the CO$_2$ NEP fluxes (Fig. 3), we still see value in presenting the component fluxes.

**I was surprised by the relative lack of direct comparisons across these three models in the discussion section. I was hoping for more explicit "unpacking" of the particular model formulations, with direct recommendations as to which model components are most justifiable given the observed data. I found that much of the discussion amounted to throw-away sentences such as line 457-460, in which little of consequence was said regarding how we should model HR.**

We thank the reviewer for this comment to clarify our discussion points and add more detail for comparison between the testbed models. The discussion was extensively modified to add depth to the comparisons being made and avoid jargon and abbreviations. We also added clarifying sentences on the scope of the analysis in the introduction section.